



# Harmonized European Union subnational crop statistics reveal climate impacts and crop cultivation shifts

Giulia Ronchetti, Luigi Nisini Scacchiafichi, Lorenzo Seguini, Iacopo Cerrani, and Marijn van der Velde

European Commission, Joint Research Centre (JRC), 21027 Ispra (VA), Italy

**Correspondence:** Giulia Ronchetti (giulia.ronchetti89@gmail.com) and Marijn van der Velde
(marijn.van-der-velde@ec.europa.eu)

**Abstract.** The availability of coherent time series of crop statistics is essential to better analyze the past, understand the present, and predict future trends in yield, area, and production. Importantly, such data also underpin assessments and subsequent policy actions that can shape future food systems that are more resilient in the face of climate change and sustainable in terms of resource use efficiency. In the European Union (EU), there currently is no legal obligation for EU countries to provide subnational crop statistics. Yet, such data could improve in-season crop forecasts, climate change impacts and adaption need assessments, and evaluation of agri-environmental schemes. The dataset described in this paper includes a harmonized collection of subnational crop statistics on area, production, and yield, collected for the EU from National Statistical Institutes (NSIs) and the Eurostat REGIO database – subnational crop statistics voluntarily contributed by EU countries. The crops considered are wheat (including soft and durum wheat), barley (including winter and spring barley), grain maize, sunflower and sugar beet. All data is harmonized towards the hierarchical structure of the Eurostat legend, and the regional classification of NUTS (Nomenclature of Territorial Units for Statistics) version 2016, to provide coherent time series. A total of 344282 records is presented here (115974 for wheat, 122705 for barley, 35274 for grain maize, 34916 for sugar beet, and 35413 for sunflower) for a total of 961 regions in 27 EU countries. Statistics are reported from 1975 to 2020, with a median time spans range of 21 years. A flagging system details for each data record information on data sources, processing steps, and quality checking results. This includes consistency checks between reported values for area, yield and production, identification of null values, missing and calculated data, information on crop legend matching, and NUTS versioning. We illustrate the value of this dataset by analyzing impacts on crops and production zone shifts in Europe due to climatic and economic factors. Recommendations and future developments of collecting subnational statistics at EU level are briefly discussed. The dataset is accessible with ECAS login at https://doi.org/10.2905/685949ff-56de-4646-a8df-844b5bb5f835 (Ronchetti et al., 2023b).

## 1 Introduction

Coherent statistics on yield, area, and production, feeds the fields of food system analysis, food security assessments (Mueller et al., 2012), and food economics, to name a few. While good quality national level crop statistics are – as a rule of thumb - available (but not always), for most countries around the world (see FAOSTAT), subnational crop statistics generally are not. Yet, coherent subnational crop statistics on yield, area, and production, are increasingly in demand by businesses, market



analysts, policymakers, scientists, and economists. Subnational statistics are published by National Statistical Institutes (NSIs) and/or Other National Authorities (ONAs) for each respective country. These subnational statistics can often be obtained through dedicated websites (e.g., United States Department of Agriculture - USDA). Several public international organizations (e.g., Eurostat and FAOSTAT) or inter-agency platforms (e.g., Agricultural Market Information System - AMIS) provide access to harmonized datasets that include statistical information on area, production and yield for the most relevant crops at national level for many countries (AMIS). Crop statistics are also provided by universities or research institutes. For instance, in 2008, Monfreda et al. (2008) provided global gridded data on harvested area and yields of 175 distinct crops around the year 2000 based on national and subnational level census statistics and a global cropland dataset. Recently, Iizumi and Sakai (2020) developed a hybrid dataset for crop yields based on agricultural census statistics and satellite remote sensing to fill temporal and spatial gaps. To support climate impact analysis, Anderson et al. (2023) published a dataset with 100-year time series of subnational wheat and maize crop statistics from global breadbaskets (Anderson et al., 2022). Yet, despite their importance, a complete and harmonized collection of subnational crop statistics for countries in the European Union (EU) currently does not exist. While Eurostat (Eurostat, 2020) receives subnational statistics from Member States (MS) and reports these in the regional database (Eurostat, a), the data provision relies on voluntary contributions, contains gaps, and does not consider changes in regional administrative boundaries through time. By assessing Eurostat regional crop statistics for the years 2016, 2017 and 2018 we found that an extended use of this dataset can be limited by some inconsistencies, such as: i) it does not contain yield data, ii) it does not report statistics for spring barley; iii) it includes records of outdated NUTS classifications (i.e. version 2006, 2010 and 2013); iv) there is an incoherent use of zero and null values for crop record entries without data; and v) aggregated production values are not coherent with the national production values published in the national database (Eurostat, a) for many combinations of countries and crops. In addition, NSIs annually provide subnational crop statistics, accounting for regional variations but also accommodating specific characteristics and needs of each individual country. A further challenge is the mapping between the different crop names, terminology, and classifications, used in the various countries. As an example, the Austrian statistical service (Statistics Austria), distinguishes spring soft wheat, winter soft wheat, spring durum wheat, and winter durum wheat, while the National Statistical Institute of Bulgaria (Republic of Bulgaria) only reports figures for total wheat, without any specifications on variety. In Spain, the Ministry of Agriculture, Fisheries and Food (MAPA), provides crop statistics on area and yield with a distinction between irrigated and non-irrigated crops.

Furthermore, from a historical and geographical point of view, the EU has changed over the years. The number of MS has changed over time, and within each country, there have been variations in terms of local subdivisions. The EU-wide Nomenclature of Territorial Units for Statistics (NUTS) includes several spatial levels for each country and has been managed by Eurostat under a series of agreements with the MS (Eurostat, 2018). These NUTS classifications can undergo changes over time (e.g. a regional unit can be merged with another region or divided, creating new regions in the process (Eurostat, b). For example, in the NUTS 2016 classification, some NUTS 2 regions in France have been recoded with respect to the NUTS 2013 classification (Eurostat, 2015b) (i.e., FR42 turned into FRF1); in Poland, a new region (i.e., PL92) has been introduced as an aggregation of regions; in Hungary, region HU10 has been split into two new regions (i.e., HU11 and HU12). These



variations could compromise the completeness of statistical time series; therefore, the reporting of crop statistics requires a
spatial reference system that is spatially consistent through time.

Given these considerations, the need for a homogeneous dataset of crop statistics that can consider and solve local variations, both from a geographical and agronomical point of view, and that is as extensive as possible, so that it can be considered as a reference for EU, becomes evident. The dataset presented in this paper is a harmonized set of subnational crop statistics for the EU[1]. Statistical values include information on area, yield and production, for the major crops cultivated in EU (Avitabile
et al., 2023). In particular, the crops considered here are: wheat (including soft and durum wheat), barley (including winter and spring barley), grain maize, sugar beet and sunflower. The harmonization follows the hierarchical structure of the Eurostat legend and is harmonized towards the administrative classification of NUTS 2016, to provide a consistent and complete dataset. The length of the time series varies by crop and subnational unit. A flagging system details information on data sources, data processing steps, and data quality. This dataset may define a benchmark for subnational crop statistics in the EU, and can be
used in agro-economic and agro-environmental studies as a reference for model calibration and validation purposes. The aim of this paper is to describe all the different steps involved in the generation of the dataset. Data collection and data processing are detailed in Section 2, while Section 3 focuses on the structure of the dataset. The value of the dataset is illustrated by mapping the lowest and highest yielding years, and by calculating shifts in crop production zones in Section 4. The paper closes with some general considerations and future perspectives on collecting subnational crop statistics in the EU.

## 2   Methods

The procedure to generate a harmonized dataset of subnational crop statistics of the EU consists of three main steps, namely i) data collection; ii) data harmonization; iii) data post-processing. The complete workflow is schematically shown in Figure 1.

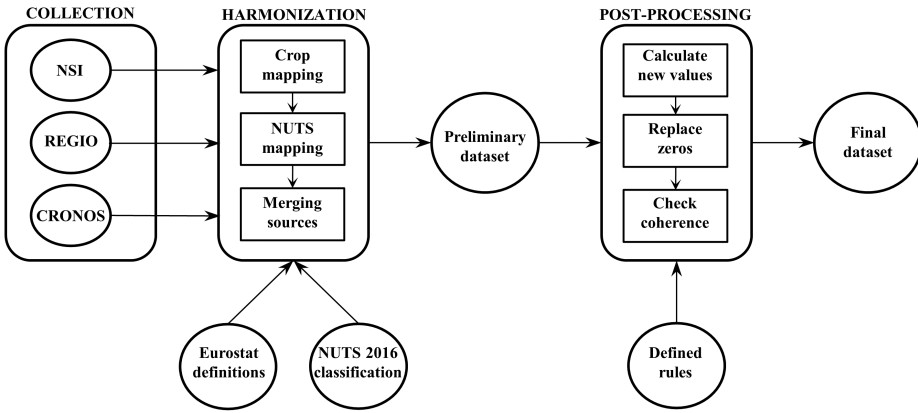

**Figure 1.** Workflow for generating the harmonized dataset of subnational crop statistics.

---

[1]In the current composition, including 27 countries.



## 2.1 Data collection

The collection of statistics on area, production, and yield for the crops considered in this dataset includes the querying of
different statistical sources, and the extraction of the required information, as far as existing and available. Data are first
collected in any provided format (i.e., text files, documents, spreadsheets, etc..), and then converted into a compatible format
to be ingested into a database. The length of statistical time series can vary depending on crops, countries and data sources, but
overall statistics cover the period from 1975 to 2020. The latest data were collected in July 2022. The main statistical sources
investigated for the data collection are the following:

- Subnational statistics from NSIs and ONAs responsible for agricultural statistics[2]: this set of agricultural statistical time
  series is obtained by directly downloading statistics from the official national websites, or by contacting the different
  NSIs or national authorities, which provide crop statistics on area, production, and yield at the lowest administrative
  level available (e.g. NUTS 3). The complete list of all the involved NSIs and national authorities is reported in Table 1,
  together with the administrative level at which the statistics are provided. From here on, this dataset is identified with the
  name "NSI".

- EUROSTAT regional database (Eurostat, a)[3]: this source consists of a database, providing figures for a list of agricultural
  commodities available for EU Member States and some neighboring countries. Data are provided at subnational level,
  namely NUTS 1 and NUTS 2 level. From here on, this dataset is identified with the name "REGIO".

- EUROSTAT national database (Eurostat, a)[4]: this source consists of a database, providing figures for a list of agricultural
  commodities available for EU Member States and some neighboring countries. Data are provided at national level,
  namely NUTS 0 level. From here on, this dataset is identified with the name "CRONOS".

## 2.2 Data harmonization

The set of regional data on crop statistics collected from EUROSTAT and NSIs is very heterogeneous. As mentioned before,
the administrative level at which statistics are provided (i.e., NUTS 1, NUTS 2 or NUTS 3) varies among the different sources,
and even within the same source, as the several statistical offices from the Member States often produce regional statistics
disaggregated at different administrative levels and levels for publication can change over the years. Moreover, the definitions
of crops are sometimes differing between national statistical services and over the years, and crop/varieties have to be properly
assigned to match reference terminology. Therefore, the statistical data collected are processed to make them comparable to
the European standards used for crop definitions and administrative units. These standards follow the convention in terms of
both aggregated name and crop definition provided by Eurostat (Eurostat, 2020). The target crops selected for this dataset are
listed in Table 2, while the selected reference layer for the administrative division is the Nomenclature of Territorial Units

---

[2]https://ec.europa.eu/eurostat/web/european-statistical-system

[3]Tables apro_cpnhr and apro_cpnhr_h

[4]Tables apro_cpnh1 and apro_cpnh1_h





**Table 1.** National source and administrative level provided of the statistics collected in the EU Member States.

| Country code | Country | Administrative level | National source |
|---|---|---|---|
| AT | Austria | NUTS 2 | Statistics Austria |
| BE | Belgium | NUTS 2 | STATBEL, Belgian statistical office |
| BG | Bulgaria | NUTS 2 | National Statistical Institute |
| CY | Cyprus | NUTS 0 | National Statistical Service |
| CZ | Czechia | NUTS 3 | Czech Statistical Office |
| DE | Germany | NUTS 3 | Federal and state statistical offices |
| DK | Denmark | NUTS 3 | Statistics Denmark |
| EE | Estonia | County | Statistics Estonia |
| EL | Greece | NUTS 3 | Hellenic Statistical Authority |
| ES | Spain | NUTS 3 | Ministry of Agriculture, Fisheries and Food |
| FI | Finland | County | Natural Resources Institute Finland |
| FR | France | NUTS 3 | Agreste, Statistical Services for Ministry of Agriculture |
| HR | Croatia | NUTS 2 | Croatian Bureau of Statistics |
| HU | Hungary | NUTS 3 | Hungarian Central Statistical Office |
| IE | Ireland | NUTS 3 | Central Statistics Office |
| IT | Italy | NUTS 3 | Italian National Institute of Statistics |
| LT | Lithuania | NUTS 3 | Official Statistics Portal |
| LU | Luxembourg | NUTS 0 | National Institute of Statistics and Economic Studies |
| LV | Latvia | NUTS 3 | Official Statistics of Latvia |
| MT | Malta | NUTS 0 | National Statistics Office |
| NL | Netherlands | NUTS 2 | Statistics Netherlands |
| PL | Poland | NUTS 2 | Statistics Poland |
| PT | Portugal | NUTS 2 | Statistics Portugal |
| RO | Romania | NUTS 3 | National Institute for Statistics |
| SE | Sweden | NUTS 3 | Statistics Sweden |
| SI | Slovenia | NUTS 3 | Republic of Slovenia Statistical Office |
| SK | Slovakia | NUTS 3 | National Institute for Statistics |

for Statistics (NUTS), version 2016 (Eurostat, 2018). Moreover, the units of measure of the collected data can vary according to the different data sources, although the standard rules suggest reporting area statistics in hectares (ha), production in tons (t) and yield in tons per hectares ($t \times ha^{-1}$). A harmonization procedure has been developed to map collected statistics to the reference hierarchy and to merge the different data sources. This procedure is fully detailed in Cerrani et al. (2023), and consists of three main steps:



**Table 2.** Eurostat definitions for the crops included in the dataset.

| Crop Name | Eurostat Code | Eurostat label | Definition |
|---|---|---|---|
| Total wheat | C1100 | Wheat and spelt | Common wheat (*Triticum aestivum L. emend. Fiori et Paol.*), spelt (*Triticum spelta L.*), einkorn wheat (*Triticum monococcum L.*) and durum wheat (*Triticum durum Desf.*). Cereal grains harvested just before maturity |
| Soft wheat | C1110 | Common wheat and spelt | Common wheat (*Triticum aestivum L. emend. Fiori et Paol.*), spelt (*Triticum spelta L.*), einkorn wheat (*Triticum monococcum L.*). Cereal grains harvested just before maturity |
| Durum wheat | C1120 | Durum wheat | Durum wheat (*Triticum durum Desf.*). Cereal grains harvested just before maturity |
| Total barley | C1300 | Barley | Barley (*Hordeum vulgare L.*). Cereal grains harvested just before maturity |
| Winter barley | C1310 | Winter barley | Barley (*Hordeum vulgare L.*) sown before or during winter. Cereal grains harvested just before maturity |
| Spring barley | C1320 | Spring barley | Barley (*Hordeum vulgare L.*) sown in the spring. Cereal grains harvested just before maturity |
| Grain maize | C1500 | Grain maize and corn-cob-mix | Maize (*Zea mays L.*) harvested for grain, as seed or as corn-cob-mix. |
| Sugar beet | R2000 | Sugar beet (excluding seed) | Sugar beet (*Beta vulgaris L.*) intended for the sugar industry, alcohol production or renewable energy production. |
| Sunflower | I1120 | Sunflower seed | Sunflower (*Helianthus annuus L.*) harvested as dry grains. |

1. Crop mapping and transformation: all crop legend classes in use by the EU Member States are mapped to match Eurostat crop legend hierarchy. Original crop classes and their mapping values per country are included as tables in the data documentation available along with the dataset (see 3.1). Any value that needed crop harmonization after the collection is marked with a proper flag.

2. NUTS mapping and transformation: any region that was affected by a NUTS version update through time is aggregated or disaggregated to coherently match NUTS version 2016 region subdivision. An algorithm was developed to compare two administrative units belonging to different NUTS versioning, and determine the regions that are equal; those which have changed only the region identifier but not their geometry; and those which have changed both geometry and identifier, providing also the weight to recompute statistics to the target layer. Any value that needed region harmonization after the collection is marked with a proper flag;





3. Data sources merging: crop and NUTS mapping and transformation are applied independently to each collected dataset. The final dataset is then generated by merging the data from the various sources from the NSIs and the Eurostat databases. The merging procedure ranks sources and first gives priority to the most recently collected data, then to data directly collected from NSIs, and finally to the regional data reported in the Eurostat database, particularly for the cases of crop statistics not reported by the NSIs. In this phase, units of measure are also homogenized, to be consistent with each other. The final dataset provides area values in hectares (ha), production values in tons (t), as well as yield values in tons per hectare ($\mathrm{t} \times \mathrm{ha}^{-1}$).

## 2.3 Data post-processing

After data harmonization, a post-processing procedure is required in order to complete the dataset with possible newly calculated values and to verify data consistency. In particular, data post-processing focuses on i) calculation of new values starting from the existing ones, both as a combination of variables and as an aggregation of crops; ii) recognition and replacement of null or erroneously zero values; iii) assessment of coherence between variables and, whenever possible, between aggregated crops.

### 2.3.1 Calculate new values

Whenever a value is missing in the dataset, a dedicated procedure tries to derive the missing value from the existing ones, through the application of simple rules. The newly calculated value can be retrieved either as a combination of variables, by exploiting the relationship between area, production, and yield values, or, in the specific case of total wheat and total barley, as an aggregation between soft and durum wheat, or between winter and spring barley, respectively. Any newly calculated value is marked with a proper flag, and, in the newly calculated record, the source is set according to the source used to calculate the new data (i.e., if both data records are from either NSIs or Eurostat then the data source for the calculated value is set accordingly, otherwise it is set as Mixed sources).

To derive a new value as a combination of variables, the formulas reported in Eq. 1, Eq. 2, and Eq. 3 are implemented in the data post-processing procedure:

$$Y = \frac{P}{A} \tag{1}$$

where $Y$ is the newly calculated value of yield in $\mathrm{t} \times \mathrm{ha}^{-1}$, $P$ is the existing value of production in t, $A$ is the existing value of area in ha, $P \geq 0$ and $A > 0$.

$$A = \frac{P}{Y} \tag{2}$$

where $A$ is the newly calculated value of area in ha, $P$ is the existing value of production in t, $Y$ is the existing value of yield in $\mathrm{t} \times \mathrm{ha}^{-1}$, $P \geq 0$ and $Y > 0$.



$$P = A \times Y \tag{3}$$

where $P$ is the newly calculated value of production in t, $A$ is the existing value of area in ha, $Y$ is the existing value of yield in $\mathrm{t} \times \mathrm{ha}^{-1}$, $A \geq 0$ and $Y \geq 0$.

The formulas reported in Eq. 1, Eq. 2, and Eq. 3 can be also applied to replace a value equal to zero or a null value. If two out

of three variables have positive values and the third one is zero or null, a new value is calculated for the third variable by means of the same equations. Moreover, if two out of three variables are zeros and the third one is null, the value for the third variable is turned into zero as well; conversely, if two out of three variables are null and the third one is equal to zero, the value for the third variable is converted in a null value.

To derive a new value of total wheat or/and total barley as an aggregation of crops, the formulas reported in Eq. 4, Eq. 5, and

160 Eq. 6 are implemented in the data post-processing procedure:

$$A_{total} = A_{crop1} + A_{crop2} \tag{4}$$

where $A_{total}$ is the newly calculated value of area in ha for total wheat (barley), $A_{crop1}$ is the existing value of area in ha for soft wheat (winter barley), $A_{crop2}$ is the existing value of area in ha for durum wheat (spring barley), $A_{crop1} > 0$, and $A_{crop2} > 0$.

$$P_{total} = P_{crop1} + P_{crop2} \tag{5}$$

where $P_{total}$ is the newly calculated value of production in t for total wheat (barley), $P_{crop1}$ is the existing value of production in t for soft wheat (winter barley), $P_{crop2}$ is the existing value of production in t for durum wheat (spring barley), $P_{crop1} > 0$, and $P_{crop2} > 0$.

$$Y_{total} = \frac{P_{total}}{A_{total}} \tag{6}$$

where $Y_{total}$ is the newly calculated value of yield in $\mathrm{t} \times \mathrm{ha}^{-1}$ for total wheat (barley), $P_{total}$ is the value of production in t for total wheat (barley), $A_{total}$ is value of area in ha for total wheat (barley), $P_{total} \geq 0$, and $A_{total} > 0$. If $P_{total}$ is not available, then the new value for yield is derived as an area-weighted average of yields. Specifically, the formula reported in Eq. 7 is applied:

$$Y_{total} = \frac{Y_{crop1} \times A_{crop1} + Y_{crop2} \times A_{crop2}}{A_{total}} \tag{7}$$

where $Y_{total}$ is the newly calculated value of yield in $\mathrm{t} \times \mathrm{ha}^{-1}$ for total wheat (barley), $Y_{crop1}$ is the existing value of yield in $\mathrm{t} \times \mathrm{ha}^{-1}$ for soft wheat (winter barley), $A_{crop1}$ is the existing value of area in ha for soft wheat (winter barley), $Y_{crop2}$ is the existing value of yield in $\mathrm{t} \times \mathrm{ha}^{-1}$ for durum wheat (spring barley), $A_{crop2}$ is the existing value of area in ha for durum wheat (spring barley), $A_{total}$ is value of area in ha for total wheat (barley), $Y_{crop1} \geq 0$, $Y_{crop2} \geq 0$, $A_{crop1} \geq 0$, $A_{crop2} \geq 0$, and $A_{total} > 0$.



### 2.3.2 Identify and replace zero with null

A dedicated procedure was developed also to identify and replace any value equal to zero that causes inconsistencies. In principle, the values of area, production and yield must be consistent with each other. Gross errors, e.g. area values equal to zero associated with positive yield values or positive production values, are detected and flagged, to provide the end-users the possibility to easily manage them. In particular, all zero values are checked and transformed into a null value should the variables disagree. Any value originally equal to zero that is replaced with a null value, is marked with a proper flag.

### 2.3.3 Check coherence

Finally, data quality control is carried out by calculating the coherence between the three variables. The threshold value for determining an inconsistency between the data is set at 1%. The coherence between statistics of area, production and yield for a given region and year is verified by applying the formula in Eq. 8:

$$|P - (A \times Y)| \leq 0.01 \times P \tag{8}$$

where $P$ is the value of production in t, $A$ is the value of area in ha, $Y$ is the value of yield in $\mathrm{t} \times \mathrm{ha}^{-1}$. Whenever the condition is met or not, a proper flag is provided. Of course, the coherence check can only be applied if none of the three variables is missing or null. The coherence between variables is calculated for all the crops included in the dataset.

Moreover, an additional coherence check is provided exclusively for statistics regarding crops wheat and barley. Since statistics for total wheat (barley) are derived by aggregation, the coherence between the statistics of the crops involved in the aggregation is also verified. In particular, the formulas reported in Eq. 9, Eq. 10, and Eq. 11 are applied. For these controls, the threshold value for determining an inconsistency between the data is also set at 1%.

$$|A_{total} - (A_{crop1} + A_{crop2})| \leq 0.01 \times A_{total} \tag{9}$$

where $A_{total}$ is the value of area in ha for total wheat (barley), $A_{crop1}$ is the value of area in ha for soft wheat (winter barley), $A_{crop2}$ is the value of area in ha for durum wheat (spring barley). Whenever the condition is met or not, a proper flag is provided. The coherence can only be checked if none of the three values is missing or null.

$$|P_{total} - (P_{crop1} + P_{crop2})| \leq 0.01 \times P_{total} \tag{10}$$

where $P_{total}$ is the value of production in t for total wheat (barley), $P_{crop1}$ is the value of production in t for soft wheat (winter barley), $P_{crop2}$ is the value of production in t for durum wheat (spring barley). Whenever the condition is met or not, a proper flag is provided. The coherence can only be checked if none of the three values is missing or null.

$$\left|Y_{total} - \frac{Y_{crop1} \times A_{crop1} + Y_{crop2} \times A_{crop2}}{A_{total}}\right| \leq 0.01 \times Y_{total} \tag{11}$$

where $Y_{total}$ is the value of yield in $\mathrm{t} \times \mathrm{ha}^{-1}$ for total wheat (barley), $Y_{crop1}$ is the value of yield in $\mathrm{t} \times \mathrm{ha}^{-1}$ for soft wheat (winter barley), $A_{crop1}$ is the value of area in ha for soft wheat (winter barley), $Y_{crop2}$ is the value of yield in $\mathrm{t} \times \mathrm{ha}^{-1}$ for





durum wheat (spring barley), $A_{crop2}$ is the existing value of area in ha for durum wheat (spring barley), $A_{total}$ is the value

of area in ha for total wheat (barley), and $A_{total} > 0$. Whenever the condition is met or not, a proper flag is provided. The coherence can only be checked if none of the values is missing or null.

## 3   Subnational crop statistics dataset

The final dataset consists of 344282 records, including 115974 records for wheat, 122705 records for barley, 35274 records for grain maize, 34916 records for sugar beet, and 35413 records for sunflower, covering 961 regions and 46 years, namely from

1975 to 2020. Table 3 presents the number of records for each crop and variable, as well as the length of each time series. In addition, details regarding the number of records for each crop and variable, the reported administrative level, and the length of each time series according to countries are summarized in Appendix A.

**Table 3.** Number of records and length of time series for each crop included in the dataset.

| Crop | First year | Last year | # records Area | # records Production | # records Yield |
|---|---|---|---|---|---|
| Total wheat | 1975 | 2020 | 13638 | 11368 | 11361 |
| Soft wheat | 1975 | 2020 | 14339 | 14201 | 19308 |
| Durum wheat | 1986 | 2020 | 10609 | 10575 | 10575 |
| Total barley | 1975 | 2020 | 14544 | 13799 | 13774 |
| Winter barley | 1975 | 2020 | 11617 | 11448 | 16796 |
| Spring barley | 1975 | 2020 | 12039 | 11878 | 16810 |
| Grain maize | 1975 | 2020 | 11777 | 11749 | 11748 |
| Sugar beet | 1975 | 2020 | 10345 | 9981 | 14590 |
| Sunflower | 1975 | 2020 | 11948 | 11765 | 11700 |

### 3.1   Structure of the dataset

The current version of the subnational crop statistics dataset is composed of 13 fields, including information on region, crop,

220   year, variable, value, source, and some additional flags. The following information can be found in the dataset:

- REGION: the code of the administrative unit which the value refers to. Administrative unit codes are based on the Eurostat classification of NUTS 2016;

- CROP_NAME: the name of the crop which the value refers to. Crop names follow the Eurostat definition (Table 2, Eurostat, 2020);

- YEAR: the year which the value refers to. Years range from 1975 to 2020, according to data availability;



- – VARIABLE: the variable which the value refers to. Specifically, the variables are Area, Production and Yield. In this dataset, the variable Area refers to area of harvesting, although not all the data sources distinguish between area of sowing and harvesting;

- – VALUE: the harmonized subnational statistics value;

- – UoM: the unit of measure for the specific value. Units of measure depend on the respective variable: Area in hectares (ha), Production in tons (t), and Yield in tons per hectares $\mathrm{t} \times \mathrm{ha}^{-1}$;

- – SOURCE: the data source of the value. Data sources are NSI, Eurostat, or Mixed (i.e., when a value is calculated from a combination of values derived from both NSI and Eurostat);

- – CALCULATED_R: flagging system, reporting if a value has been derived from a NUTS version different from NUTS 2016, by means of NUTS mapping and transformation procedure (Section 2.2);

- – CALCULATED_C: flagging system, reporting if a value has been derived from a combination of crops to match Eurostat definitions, by means of crop mapping and transformation procedure (Section 2.2);

- – CALCULATED_V: flagging system, reporting if a value, that originally was missing or null or zero, has been calculated during the post-processing phase, by means of the Equations presented in Section 2.3.1;

- – ZERO_AS_NULL: flagging system, reporting if a value of zero has been turned into a null value during the post-processing phase because of any inconsistencies, as described in Section 2.3.2;

- – COHERENCE_APY: flagging system, reporting if there is agreement among values of Area, Production and Yield for the same region, crop and year, according to the rule defined in Eq. 8;

- – COHERENCE_CROP: flagging system, reporting if there is agreement among values of Total wheat (Total barley), Soft wheat (Winter barley), and Durum wheat (Spring barley) for the same region, variable and year, according to the rules defined in Eq.9, Eq.10, and Eq.11.

## 3.2 Flagging system

A total of six flags are reported together with the data, representing additional information on data processing and data quality. This flagging system can help the users of this dataset to have a clear knowledge of the originality and the level of processing underlying the values they are dealing with, as well as to verify the reliability of the data. The flags *CALCULATED_R*, *CALCU-LATED_C*, *CALCULATED_V*, and *ZERO_AS_NULL* are set as *Yes* or left blank, depending on whether the specific condition is met or not. The flags about coherence, namely *COHERENCE_APY*, and *COHERENCE_CROP* are set as *Yes* or *No*, depending on whether coherence is verified or not, and left blank when it is not possible to evaluate coherence due to missing or null values, as described in Section 2.3.3. Maps in Figure 2 represent regions whose values were derived from a transformation of NUTS and/or crops. The need of deriving values from a transformation of regions arises from two main reasons: data sources





providing values according to a different NUTS version classification prior to version 2016, and/or data sources providing values according to their own internal administrative subdivision. The former is the case of regions in Ireland, Italy, and Poland, where the less recent crop statistics values were updated to match NUTS classification version 2016, as data sources originally provided them according to a different NUTS versioning. The latter is the case of regions in Denmark, Estonia, and Finland,

whose NSIs report crop statistics values using their own internal administrative units (e.g., county) not compliant (i.e., Estonia) or only partially compliant (i.e., Denmark and Finland) with NUTS classification. Finally, crop statistics values for Greece were derived from a NUTS transformation, as regions in Greece have deeply changed after NUTS classification version 2010 but also NSI reports detailed crop statistics for small islands that need to be grouped to match the NUTS classification in use.

As regards values calculated from a transformation of crops, wheat and grain maize have required the highest number of

265 transformations. Most NSIs reports statistics for wheat with a distinction between winter and spring varieties, and for maize with details about seeds and corn-cob mix and/or about irrigation. The map of Figure 2b shows that most countries required computing a transformation of crops only for one crop (i.e. soft wheat), while in France and Romania we derived values for two crops (i.e. soft and durum wheat). Only in Austria, almost all crops (i.e. total wheat, soft wheat, durum wheat and grain maize) required to be harmonized with Eurostat crop definitions. Also, not all the data sources reported clear and coherent details how they account for corn-cob mix in their publications. In Figure 3, maps show the shares of records with verified

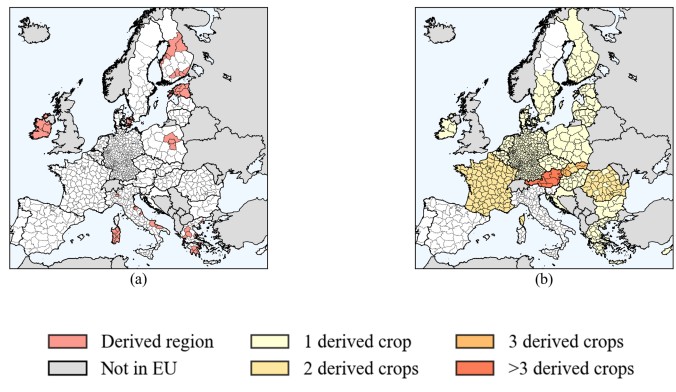

**Figure 2.** Regions whose the reported values were derived from a transformation of NUTS (a) and/or crops (b) for at least one crop included in this dataset.

coherence among variables for each region and crop, as well as the maps of Figure 4 and 5 display the shares of records with verified coherence for total wheat and total barley, respectively. The shares of records with verified coherence among variables and/or crops are computed by comparing the number of coherent records (i.e. Yes flags in the coherence columns) with the total number of records in the time series for each crop and region. A share close to 100% means that coherence is verified in

all the reported records, while a share close to 0% represents very few records with verified coherence in the time series.

From the maps of Figure 3 it is evident that coherence rate among variables is very high for all regions and crops with a few exceptions, including Germany and Finland. For Germany, the low coherence rate is due to the fact that time series of variable



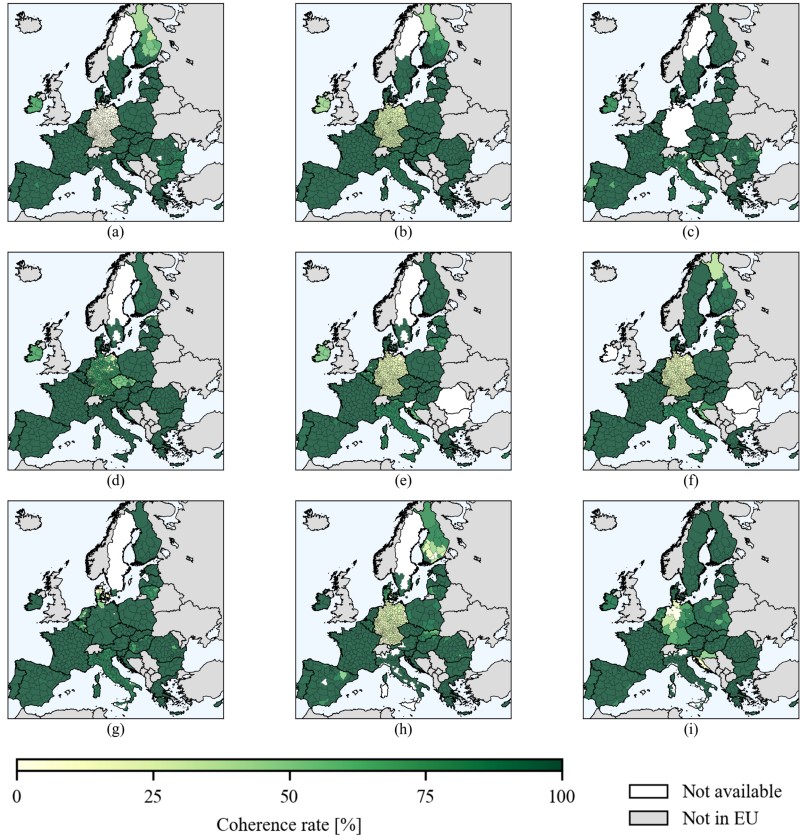

**Figure 3.** The shares of records with verified coherence among area, production and yield variables for each region and crop: a) Total wheat, b) Soft wheat, c) Durum wheat, d) Total barley, e) Winter barley, f) Spring barley, g) Grain maize, h) Sugar beet, i) Sunflower.

yield are more complete and longer than the ones of area and production, therefore coherence checks can be computed only for few records. Differently, for Finland, the coherence check rule is not verified in the northernmost regions with low agricultural activity where crop statistics are affected by approximation errors.

As regarding coherence among total, soft and durum wheat (Figure 4), coherence rate values differ from country to country and mostly depend from the availability of durum wheat statistics. Coherence rates are high for regions in southern Europe, where most durum wheat is produced, and in regions where the production of durum wheat is absent, while rates are low in regions where time series of durum wheat statistics are discontinuous or rarely published. Similarly, coherence rates among total, winter and spring barley vary from country to country, as some NSIs do not provide distinct statistics for winter and spring wheat, as for the case of Romania and Bulgaria. Finally, in Germany, coherence rates of yield are lower than the ones of area and production, because of the different lengths of these time series preventing to apply the coherence checks formula (Eq. 11).





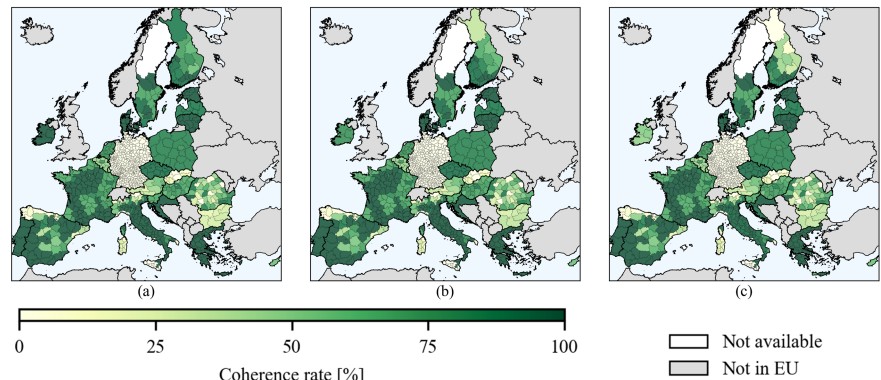

**Figure 4.** The shares of records with verified coherence among total, soft and durum wheat for each region and variable: a) Area, b) Production, c) Yield.

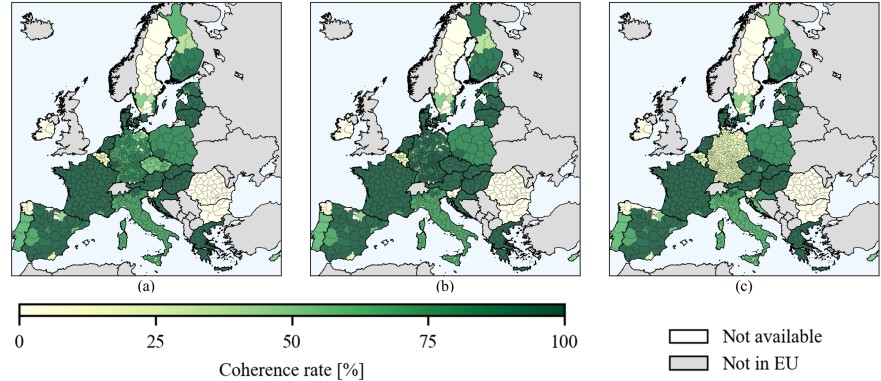

**Figure 5.** The shares of records with verified coherence among total, winter and spring barley for each region and variable: a) Area, b) Production, c) Yield.

## 4 Discussion

### 4.1 Potential uses of this dataset

In order to highlight the novelty and significance of this dataset, some potential uses are presented below and discussed. The reported analyses exploit the major strengths of this dataset, including the fine spatial resolution of data, the length, completeness and coherence of time series.



### 4.1.1 Lowest and highest yielding years

Crop statistics included in this dataset cover a long time range, from 1975 to 2020, to allow time series analysis. In this study, we performed a simple analysis, selecting for each region and crop the year when the lowest and highest yield values occurred. Results for soft wheat, grain maize, and sugar beet are reported in Figure 6. No detrending procedure was computed on yield values. Therefore, in regions where an important trend component related to the improvement of agro-management practices exists (Ronchetti et al., 2023a; García-Condado et al., 2019; Ceglar et al., 2016; Finger, 2010), the highest yield values were

observed (Figure 6 right) in the most recent years (i.e. from 2016 to 2020), while the lowest yield values were registered at the beginning of the time series, mostly in the early 2000s (Figure 6 left). This is particularly evident in eastern and northeastern EU countries, including Bulgaria, Romania, Hungary, Slovakia, Poland and Lithuania. In the other MS, improvements in agronomic techniques and management practices have less impact on crop yields, which are more dependent on climate and weather conditions within the season. Trend effects on yield values are low, and in these countries inter-annual variability of

yields is high. Yet, the years when the highest and lowest yield values occurred are heterogeneous, vary from region to region and cover the whole time range. The maps on the left of Figure 6 can also reveal extreme years, whose unfavourable conditions have affected crops and led to low yield values. The most outstanding is year 2016 in France for soft wheat (Figure 6a), when a combination of factors caused the most severe yield loss in over half a century in one of the leading wheat-producing regions of Europe (Nóia Júnior et al., 2023; Ben-Ari et al., 2018). This exceptional yield decline is well depicted in the map, with dark red

colours extending throughout the north-easternmost administrative units of France. Similarly, in 2018 a severe summer drought affected summer crops yields in central and eastern Europe (Beillouin et al., 2020; Webber et al., 2020). As a consequence, 2018 resulted the year when the lowest yield values for grain maize occurred in Germany and Belgium, as represented in Figure 6c. Furthermore, these maps show that the distribution of yields does not necessarily follow national boundaries, but often there are clusters of regions with similar behaviour in neighbouring states. Weather conditions, soil types, agronomic

practices, but also historical factors, have determined clusters of agricultural regions that do not coincide with national borders (Guth and Smędzik-Ambroży, 2020; Guiomar et al., 2018; Reiff et al., 2018). Regions in Poland are the most evident case: western regions tend to uniform with eastern German regions, whereas eastern Polish regions create an agronomic cluster with the Baltics.

### 4.1.2 Crop production zones and shifts

In this dataset, crop statistics are reported at the finest spatial detail available, namely subnational administrative distribution. This allows to perform analyses and comparisons within each country and assess the inter-country spatial distribution of crop production. Figure 7 shows the spatial distribution of crop production centroids within each country in the time period 2000 - 2020. To produce the map, we first selected production statistics for the period of interest and compute the average for each subnational unit, then we extracted geometric centroids for each subnational unit, finally crop production centroids for

each country were calculated as the weighted aggregation of subnational centroids using production averages as weights. The resulting map represents the spatial distribution of the production for each crop in the different countries. In some countries,



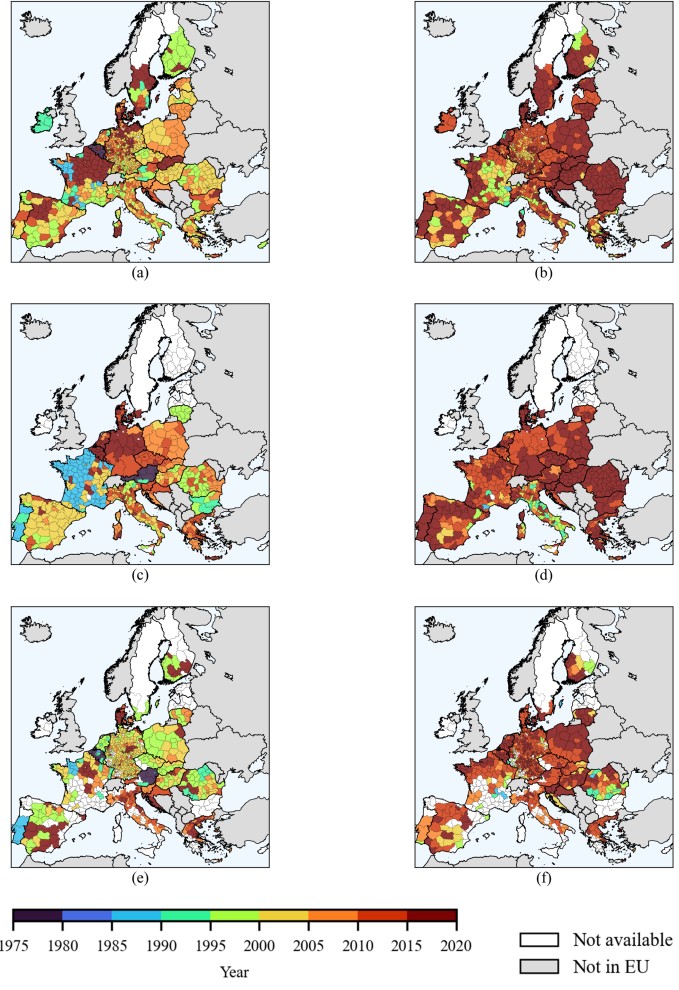

**Figure 6.** Years when the lowest (left) and the highest (right) yield values occurred for each region for crop *Soft wheat* (a, b), *Grain maize* (c, d), *Sugar beet* (e, f).

including central and northeastern Europe (e.g. Czechia, Slovakia, Poland, Lithuania, Latvia, Estonia) production centroids for the different crops are located in the same area, roughly corresponding to the geometric centroids of the country. This highlights that crop production is homogeneously distributed in the different regions of these countries and all subnational units contribute almost equally to the production of each crop (Joint Research Centre, 2023; Lennert and Farkas, 2020; Rega et al., 2020; López-Lozano et al., 2015). In Finland, Denmark, Sweden and Austria, the map points out the presence of agricultural active regions, as production centroids are centred in a small area that do not correspond to the geometric centroids (Peltonen-Sainio and Jauhiainen, 2020; Jørgensen et al., 2019; Piikki and Söderström, 2019; Stürmer et al., 2013). In the remaining parts of these countries, agriculture activities are limited by mountains, forests and non-favourable climatic conditions. Finally, in

wide and/or north-south oriented European countries, including Spain, France, Italy, Germany, and Romania, crop production centroids are widespread over the country with crops located in their most productive regions in each country (Ballot et al., 2022; Schmitt et al., 2022; d'Andrimont et al., 2021; Ribeiro et al., 2020). As an example, in Italy, the durum wheat centroid is located in the south, while grain maize centroid is located in the north in the middle of Pianura Padana; in Spain, all crops centroids are distributed around Castilla y Leon region, with the exception of durum wheat centroid which is shifted in the

south close to Andalusia region; in France, centroids for soft wheat, barley and sugar beet are located in the north-east part of the country, while centroids for grain maize, sunflower and durum wheat are placed in western/southwestern regions.

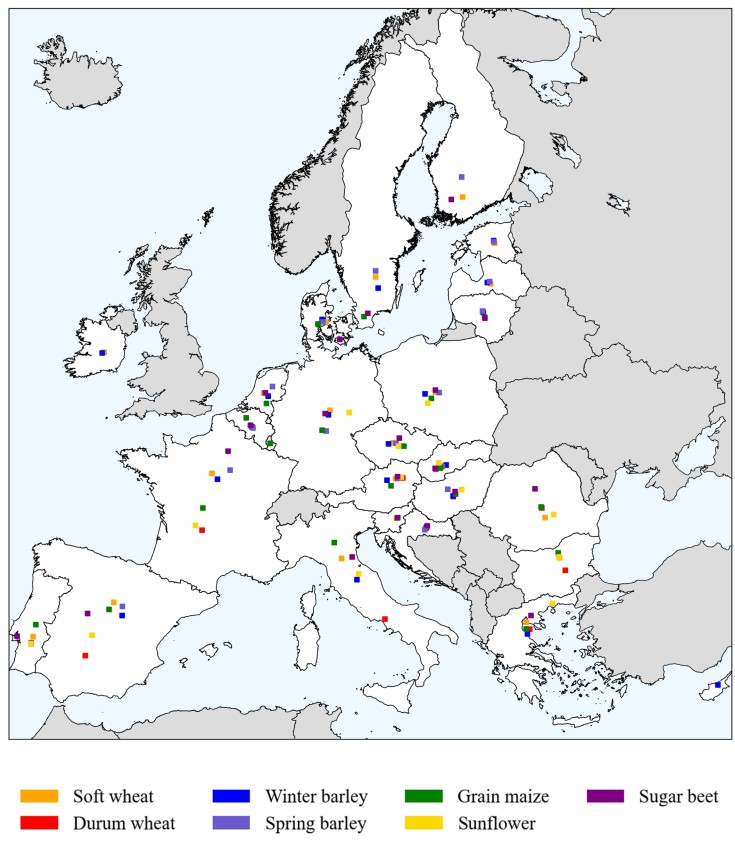

**Figure 7.** Spatial representation of crop production centroids within each country in the time period 2000 - 2020.

By exploiting both the fine spatial resolution of data and the length and completeness of time series, we performed an analysis on the distribution of crop production zones in the EU and their evolution and changes in time. First, we divided the dataset in two parts: the first one including statistics from 2000 to 2009, the second one with statistics from 2010 to 2019. Then, following

the approach above, we calculated the average production in both time periods and computed crop production centroids for the EU for both time periods, as weighted average of each regional centroid, using crop production averages as weights. Finally, we





compared the two centroids and generated the displacement vectors as the difference between crop production centroids of the second and first time period. Results are displayed in the polar plot of Figure 8. These vectors represent the shift of production in EU in a 10 years time frame for each crop. Overall, the prevalent directions of vectors are from northwest to east, testifying

an increasing contribution of northern and eastern EU MS to European agricultural production. The lowest shifts (i.e. rate equal to 5 km per year) are reported for spring barley, winter barley and sugar beet, mostly northward. The greatest displacements are observed for sunflower, followed by durum wheat, grain maize and soft wheat. For sunflower, the production shift follows a rate of 25 km per year, in the eastern direction, due to continuously growing production in Romania, Hungary and Bulgaria. These countries jointly accounted for 70% of the total European sunflower production in 2019 while in 2009 for almost 53%.

Likewise, the vector for soft wheat points towards an eastern direction, at a rate of 10 km per year. For soft wheat, an increasing contribution of eastern and north-eastern European states, such as Romania and Poland, but also the Baltics, is highlighted. Similarly, Sloat et al. (2020) found a northward migration for wheat in Eastern European countries. The production centroids of grain maize and durum wheat are also displaced by nearly 10 km per year but northeastward. These crops, typically cultivated in southern Europe, including Spain, Italy and France, are expanding more and more northeastward. Climate change is one

of the causes of these shifts. Ceglar et al. (2019) pointed out that agro-climate zones are migrating northward and that the migrating rate is accelerating due to climate change. Crop production may shift northward because of larger suitability and more favourable climate conditions, while in southern Europe adverse conditions may affect crop production. In this scenario, southern regions, including the Mediterranean area, may lose suitability to grow specific crops in favor of northern European regions (Ceglar et al., 2019; Fontana et al., 2015). Ceglar et al. observed a migration velocity of agro-climate zones northward of

100 km per 10 years solely using agro-metereological indicators, we find similar results through the analysis of crop statistics. In addition, production shifts not mediated by climate but rather by economic opportunity, largely in eastern Europe, also occur, as illustrated by the high eastward shift rate of sunflower production. Hence, the completeness of subnational crop statistics here presented may be of help to reveal agro-climate zones shifts and crop production impacts.

## 4.2 Outlook and recommendations

In the EU, a new framework regulation governing the collection of Statistics on Agricultural Inputs and Outputs (SAIO) will apply from 1st January, 2025. The collection of subnational crop statistics will become legally binding (European Parliament, EPRS). In SAIO, MS will have to report crop statistics on area and production before 30th September the year after (N+1). Guidance is given on how the reporting should be done, e.g. in terms of clean, dry weight of grains at the standard market humidity level in the country (Eurostat, 2015a). National standard humidity level needs to be reported too for possible re-

calculations to standardized EU values. For sugar beet, MS will have to provide data on the sugar content of the harvested production.

This new regulation should improve the availability and quality of subnational statistics at EU-level considerably. Nevertheless, a few practical recommendations have been identified during the progress of this study, which we list here:



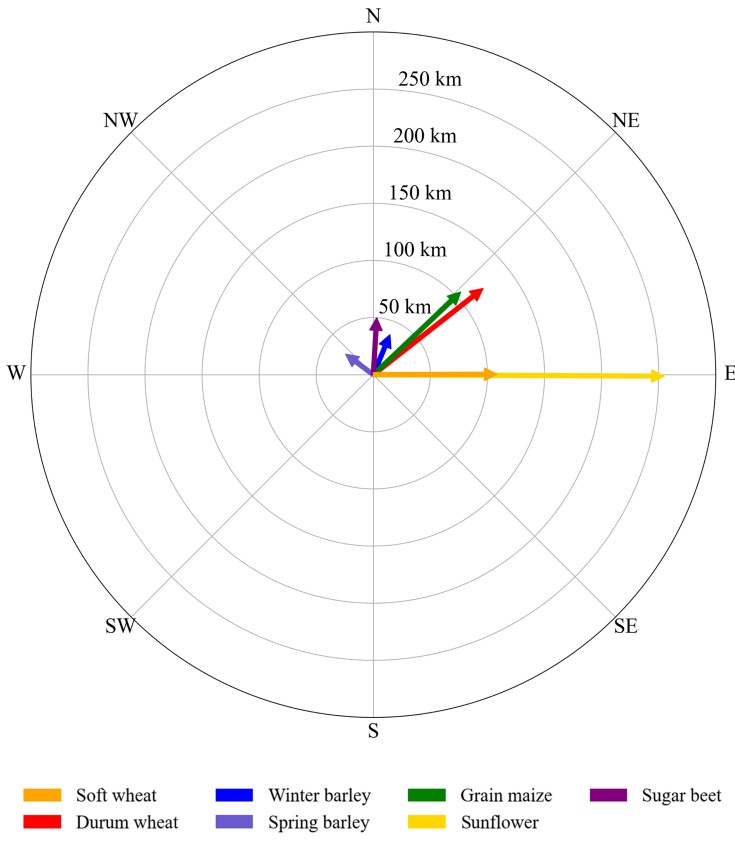

**Figure 8.** Displacement vectors representing the shifts of production centroids for each crop in EU, from the first to the second decade of the 2000s

1. *Mapping to a common legend.* There is no integral publication available on how MS match their national crop legends in national languages in their reporting to the harmonized Eurostat crop legend (Eurostat, 2020). While several newer MS have adopted their national reporting to the Eurostat crop legend, older MS have not. Since all MS must go through this exercise, this information should exist and the availability of such a document would be very valuable. We provide such a (re-engineered) list in the ancillary file Mapping_eurostat_legend.xlsx, included in our data repository (see Section 5).

2. *Adherence to the latest NUTS version.* In reporting to Eurostat, MS should use the latest and most up to date NUTS version.

3. *Clarity on reported data values.* In SAIO, clarity has been created on the definition of area, and sown area is the area data that will be collected, and this is often sourced from the farm holding declarations in the MS IACS (Integrated Administration and Control System). We find that incoherence between area, production, and yield often arises from the fact that NSIs use harvested areas to calculate production and yield. In cases with those inconsistencies, the sown area is





then not updated. Therefore, depending on the methodology MS use to calculate harvested production, clarity should be
        provided on whether harvested area is used as part of the calculation.

   4. *Sharing common parameters.* Reporting and using of unequivocal standards on humidity, and if relevant on oil and sugar
      content.

   5. *Updating past statistical data values.* In the creation of this dataset, we also identified inconsistencies with respect to
past statistics on area, production, and yiled. MS should go through an effort to update the past statistical time series
      available at Eurostat. While SAIO will cover data from 2025 and onward, ideally, such an update would be an integrated
      exercise since the availability of such time series will underpin our capacity to improve the assessment and forecasting of
      the impact of extreme weather. Such information is essential for global forecasting systems, as the case of the European
      Commission's Joint Research Centre (EC-JRC) MARS (Monitoring Agricultural Resources) Crop Yield Forecasting
System (MCYFS, Van der Velde et al. (2019)).

   6. *Promoting the use of automated data flows.* While sourcing and reporting of crop statistics from MS to Eurostat has
      significantly improved in the last years, with standardized online forms facilitating this, ample room exist for use of au-
      tomated data flows. For instance, if each MS keeps a registry with a database of national and subnational crop statistics
      following standardized metadata, an application programming interface (API) could automatically source this informa-
tion from NSIs websites. This would improve the transparency and timeliness of reporting of crop statistics considerably,
      including for national and preliminary statistics.

## 5   Data availability

The harmonized dataset of subnational crop statistics presented in this paper is available for download at https://doi.org/10.
2905/685949ff-56de-4646-a8df-844b5bb5f835 (Ronchetti et al., 2023b), accessible with ECAS login. The data publication
includes ancillary documentation along with the dataset to provide users with useful information for a deep understand-
ing of the dataset and which can be found here: https://agri4cast.jrc.ec.europa.eu/DataPortal/Resource_Files/SupportFiles/36/
Allmetadata.zip. The whole publication contains the following files:

   – AllCrops_subnstats_2023.csv: crop statistics dataset stored in CSV format. The complete list of attributes and fields
     included in the dataset are reported in Section 3;

– ResourceInfo.pdf: document reporting a description of the whole data publication;

   – Regional db Structure and Flagging system.pdf: document reporting a description of the structure of the dataset and
     details of the associated flagging system;

   – Summary of algorithm for disaggregation.pdf: document reporting a short summary of the procedures used for data
     harmonization;




- Mapping_eurostat_legend.xlsx: excel file, including a set of tables reporting original crop classes in original language
and their mapping to the common Eurostat legend (Table 2) per country;

- Country_fiches.zip: compressed files containing a set of tables with summary statistics per MS and crop, as well as
Supplementary Figures mapping the number of records across the EU, the length of time series, for each of the area,
yield, and production crop statistics for each MS.

For the review process, anonymous access to the dataset and all metadata is guarantee with this link: https://jeodpp.jrc.ec.
europa.eu/ftp/private/zyWXpm2b1/6oD5jx7UjaibM53u/Subnational_crop_statistics/

## 6 Conclusions

This data paper presents a subnational dataset of crop statistics for major crops in EU. The dataset includes harmonized
subnational crop statistics on area, production, and yield, collected for the EU from National Statistical Institutes and the
430 Eurostat REGIO database. Crop statistics are available for soft, durum, and total wheat, winter, spring, and total barley, grain
maize, sunflower, and sugar beet, for a total of 344282 reported values. A dedicated flagging system has been set for the dataset,
to provide users with more information on data quality and coherence.

The dataset requires frequent activities dedicated to maintaining and updating it, including efforts to provide complete time
series at the finest available subnational level and for an increasing number of crops. Nonetheless, at the time being, this
dataset can be considered as a benchmark for subnational crop statistics in Europe, and can serve as a reference for setting
methodologies and indicators, including calibration and validation of agronomic models and crop yield forecasting systems,
and many research studies will benefit of it. Among the potential uses of this dataset, in this paper we presented the effects of
climate change on crop production by analyzing crop statistics only.




# Appendix A: Overview of data records

Number of records included in the dataset, length of time series and reported administrative level for each crop.

**Table A1.** Number of records included in the dataset, length of time series and reported administrative level for crop *Total wheat*.

| Country | NUTS level | First year | Last year | # records Area | # records Production | # records Yield |
|---|---|---|---|---|---|---|
| AT | NUTS 2 | 1975 | 2020 | 351 | 352 | 351 |
| BE | NUTS 2 | 1975 | 2020 | 466 | 469 | 465 |
| BG | NUTS 2 | 1991 | 2020 | 180 | 180 | 180 |
| CY | NUTS 0 | 1987 | 2020 | 34 | 34 | 34 |
| CZ | NUTS 3 | 1998 | 2020 | 322 | 322 | 322 |
| DE | NUTS 3 | 1999 | 2020 | 2198 | 0 | 0 |
| DK | NUTS 3 | 2006 | 2020 | 165 | 165 | 165 |
| EE | NUTS 3 | 2004 | 2020 | 85 | 85 | 85 |
| EL | NUTS 3 | 1998 | 2019 | 1047 | 1047 | 1047 |
| ES | NUTS 3 | 1998 | 2020 | 1069 | 1066 | 1066 |
| FI | NUTS 3 | 1998 | 2020 | 352 | 334 | 333 |
| FR | NUTS 3 | 1989 | 2020 | 2615 | 2606 | 2606 |
| HR | NUTS 2 | 2008 | 2020 | 26 | 26 | 26 |
| HU | NUTS 3 | 1996 | 2020 | 490 | 490 | 490 |
| IE | NUTS 2 | 1990 | 2020 | 93 | 63 | 63 |
| IT | NUTS 3 | 1995 | 2020 | 1922 | 1919 | 1919 |
| LT | NUTS 3 | 2000 | 2020 | 209 | 209 | 209 |
| LU | NUTS 0 | 1975 | 2020 | 46 | 46 | 46 |
| LV | NUTS 3 | 2000 | 2018 | 95 | 95 | 95 |
| MT | NUTS 0 | 2000 | 2020 | 21 | 21 | 21 |
| NL | NUTS 2 | 1994 | 2020 | 324 | 324 | 324 |
| PL | NUTS 2 | 1999 | 2020 | 369 | 369 | 369 |
| PT | NUTS 2 | 1986 | 2020 | 245 | 245 | 245 |
| RO | NUTS 3 | 1998 | 2020 | 469 | 456 | 455 |
| SE | NUTS 3 | 2000 | 2020 | 327 | 327 | 327 |
| SI | NUTS 2 | 2007 | 2020 | 28 | 28 | 28 |
| SK | NUTS 3 | 2007 | 2018 | 90 | 90 | 90 |



**Table A2.** Number of records included in the dataset, length of time series and reported administrative level for crop *Soft wheat*.

| Country | NUTS level | First year | Last year | # records Area | # records Production | # records Yield |
|---|---|---|---|---|---|---|
| AT | NUTS 2 | 1995 | 2020 | 234 | 234 | 234 |
| BE | NUTS 2 | 1975 | 2020 | 439 | 438 | 438 |
| BG | NUTS 2 | 1995 | 2020 | 65 | 65 | 65 |
| CY | NUTS 0 | 2000 | 2020 | 21 | 21 | 21 |
| CZ | NUTS 3 | 1998 | 2020 | 322 | 322 | 322 |
| DE | NUTS 3 | 1999 | 2020 | 1956 | 1868 | 6973 |
| DK | NUTS 3 | 2006 | 2020 | 165 | 165 | 165 |
| EE | NUTS 3 | 2004 | 2020 | 85 | 85 | 85 |
| EL | NUTS 3 | 1998 | 2019 | 1047 | 1047 | 1047 |
| ES | NUTS 3 | 1998 | 2020 | 1108 | 1108 | 1108 |
| FI | NUTS 3 | 1998 | 2020 | 353 | 334 | 334 |
| FR | NUTS 3 | 1989 | 2020 | 3028 | 3028 | 3028 |
| HR | NUTS 2 | 2008 | 2020 | 26 | 26 | 26 |
| HU | NUTS 3 | 2002 | 2020 | 380 | 380 | 380 |
| IE | NUTS 2 | 1990 | 2020 | 93 | 63 | 63 |
| IT | NUTS 3 | 1995 | 2020 | 2325 | 2325 | 2328 |
| LT | NUTS 3 | 2000 | 2020 | 209 | 209 | 209 |
| LU | NUTS 0 | 1975 | 2020 | 46 | 46 | 46 |
| LV | NUTS 3 | 1997 | 2018 | 110 | 110 | 110 |
| MT | NUTS 0 | 2000 | 2020 | 21 | 21 | 21 |
| NL | NUTS 2 | 1994 | 2020 | 324 | 324 | 324 |
| PL | NUTS 2 | 2003 | 2020 | 301 | 301 | 301 |
| PT | NUTS 2 | 1986 | 2020 | 245 | 245 | 245 |
| RO | NUTS 3 | 1998 | 2020 | 955 | 954 | 954 |
| SE | NUTS 3 | 1990 | 2020 | 437 | 438 | 437 |
| SI | NUTS 2 | 2007 | 2020 | 28 | 28 | 28 |
| SK | NUTS 3 | 2017 | 2018 | 16 | 16 | 16 |



**Table A3.** Number of records included in the dataset, length of time series and reported administrative level for crop *Durum wheat*.

| Country | NUTS level | First year | Last year | # records Area | # records Production | # records Yield |
|---|---|---|---|---|---|---|
| AT | NUTS 2 | 1995 | 2020 | 171 | 172 | 171 |
| BE | NUTS 2 | 2000 | 2020 | 227 | 227 | 227 |
| BG | NUTS 2 | 1998 | 2020 | 60 | 60 | 60 |
| CY | NUTS 0 | 1987 | 2020 | 34 | 34 | 34 |
| CZ | NUTS 3 | 2000 | 2020 | 294 | 294 | 294 |
| DE | - | - | - | - | - | - |
| DK | NUTS 3 | 2006 | 2020 | 165 | 165 | 165 |
| EE | NUTS 3 | 2004 | 2020 | 85 | 85 | 85 |
| EL | NUTS 3 | 1998 | 2019 | 1047 | 1047 | 1047 |
| ES | NUTS 3 | 1998 | 2020 | 735 | 732 | 732 |
| FI | NUTS 3 | 2000 | 2020 | 324 | 324 | 324 |
| FR | NUTS 3 | 1989 | 2020 | 2615 | 2608 | 2608 |
| HR | NUTS 2 | 2008 | 2020 | 26 | 26 | 26 |
| HU | NUTS 3 | 1998 | 2020 | 397 | 397 | 399 |
| IE | NUTS 2 | 1990 | 2020 | 93 | 84 | 84 |
| IT | NUTS 3 | 1995 | 2020 | 2344 | 2341 | 2341 |
| LT | NUTS 3 | 2000 | 2020 | 209 | 209 | 209 |
| LU | NUTS 0 | 2000 | 2020 | 21 | 21 | 21 |
| LV | NUTS 3 | 2000 | 2018 | 95 | 95 | 95 |
| MT | NUTS 0 | 2000 | 2020 | 21 | 21 | 21 |
| NL | NUTS 2 | 2000 | 2020 | 252 | 252 | 252 |
| PL | NUTS 2 | 2003 | 2020 | 301 | 301 | 301 |
| PT | NUTS 2 | 1986 | 2020 | 245 | 245 | 245 |
| RO | NUTS 3 | 1998 | 2020 | 469 | 456 | 455 |
| SE | NUTS 3 | 2000 | 2020 | 327 | 327 | 327 |
| SI | NUTS 2 | 2000 | 2020 | 42 | 42 | 42 |
| SK | NUTS 3 | 2017 | 2018 | 10 | 10 | 10 |



**Table A4.** Number of records included in the dataset, length of time series and reported administrative level for crop *Total barley*.

| Country | NUTS level | First year | Last year | # records Area | # records Production | # records Yield |
|---|---|---|---|---|---|---|
| AT | NUTS 2 | 1975 | 2020 | 413 | 413 | 413 |
| BE | NUTS 2 | 1975 | 2020 | 468 | 470 | 446 |
| BG | NUTS 2 | 1991 | 2020 | 180 | 180 | 180 |
| CY | NUTS 0 | 1987 | 2020 | 34 | 34 | 34 |
| CZ | NUTS 3 | 1998 | 2020 | 322 | 322 | 322 |
| DE | NUTS 3 | 1999 | 2020 | 2207 | 1491 | 1491 |
| DK | NUTS 3 | 2006 | 2020 | 165 | 165 | 165 |
| EE | NUTS 3 | 2004 | 2020 | 83 | 84 | 83 |
| EL | NUTS 3 | 1998 | 2019 | 1047 | 1047 | 1047 |
| ES | NUTS 3 | 1998 | 2020 | 1008 | 1008 | 1008 |
| FI | NUTS 3 | 1998 | 2020 | 354 | 354 | 354 |
| FR | NUTS 3 | 1989 | 2020 | 3027 | 3027 | 3027 |
| HR | NUTS 2 | 2008 | 2020 | 26 | 26 | 26 |
| HU | NUTS 3 | 1996 | 2020 | 492 | 492 | 492 |
| IE | NUTS 2 | 1990 | 2020 | 93 | 63 | 63 |
| IT | NUTS 3 | 2000 | 2020 | 2122 | 2121 | 2121 |
| LT | NUTS 3 | 2000 | 2020 | 208 | 208 | 208 |
| LU | NUTS 0 | 1975 | 2020 | 46 | 46 | 46 |
| LV | NUTS 3 | 1997 | 2018 | 110 | 110 | 110 |
| MT | NUTS 0 | 2000 | 2020 | 21 | 21 | 21 |
| NL | NUTS 2 | 1994 | 2020 | 323 | 323 | 323 |
| PL | NUTS 2 | 1999 | 2020 | 365 | 365 | 365 |
| PT | NUTS 2 | 1986 | 2020 | 245 | 245 | 245 |
| RO | NUTS 3 | 1998 | 2020 | 957 | 956 | 956 |
| SE | NUTS 3 | 1995 | 2020 | 104 | 104 | 104 |
| SI | NUTS 2 | 2007 | 2020 | 28 | 28 | 28 |
| SK | NUTS 3 | 2007 | 2018 | 96 | 96 | 96 |





**Table A5.** Number of records included in the dataset, length of time series and reported administrative level for crop *Winter barley*.

| Country | NUTS level | First year | Last year | # records Area | # records Production | # records Yield |
|---------|-----------|-----------|-----------|----------------|----------------------|-----------------|
| AT | NUTS 2 | 1975 | 2020 | 413 | 413 | 413 |
| BE | NUTS 2 | 2009 | 2020 | 126 | 126 | 125 |
| BG | - | - | - | - | - | - |
| CY | NUTS 0 | 2004 | 2020 | 17 | 17 | 17 |
| CZ | NUTS 3 | 1998 | 2020 | 322 | 322 | 322 |
| DE | NUTS 3 | 1999 | 2020 | 1752 | 1624 | 7075 |
| DK | NUTS 3 | 2006 | 2020 | 165 | 165 | 165 |
| EE | NUTS 3 | 2004 | 2020 | 77 | 72 | 72 |
| EL | NUTS 3 | 1998 | 2019 | 1047 | 1047 | 1047 |
| ES | NUTS 3 | 1998 | 2020 | 901 | 902 | 901 |
| FI | NUTS 3 | 2000 | 2020 | 314 | 314 | 314 |
| FR | NUTS 3 | 1989 | 2020 | 3027 | 3027 | 3027 |
| HR | NUTS 2 | 2008 | 2020 | 26 | 26 | 26 |
| HU | NUTS 3 | 1996 | 2020 | 492 | 492 | 492 |
| IE | NUTS 2 | 2000 | 2020 | 63 | 33 | 33 |
| IT | NUTS 3 | 2005 | 2020 | 1512 | 1512 | 1409 |
| LT | NUTS 3 | 2000 | 2020 | 208 | 208 | 210 |
| LU | NUTS 0 | 1975 | 2020 | 46 | 46 | 46 |
| LV | NUTS 3 | 1997 | 2018 | 110 | 110 | 110 |
| MT | NUTS 0 | 2000 | 2020 | 21 | 21 | 21 |
| NL | NUTS 2 | 1994 | 2020 | 323 | 323 | 323 |
| PL | NUTS 2 | 2003 | 2020 | 301 | 301 | 301 |
| PT | NUTS 2 | 1999 | 2020 | 154 | 147 | 147 |
| RO | - | - | - | - | - | - |
| SE | NUTS 3 | 1995 | 2020 | 104 | 104 | 104 |
| SI | - | - | - | - | - | - |
| SK | NUTS 3 | 2007 | 2018 | 96 | 96 | 96 |

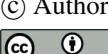



**Table A6.** Number of records included in the dataset, length of time series and reported administrative level for crop *Spring barley*.

| Country | NUTS level | First year | Last year | # records Area | # records Production | # records Yield |
|---|---|---|---|---|---|---|
| AT | NUTS 2 | 1975 | 2020 | 414 | 414 | 414 |
| BE | NUTS 2 | 2009 | 2020 | 120 | 120 | 120 |
| BG | - | - | - | - | - | - |
| CY | NUTS 0 | 2004 | 2020 | 17 | 17 | 17 |
| CZ | NUTS 3 | 1998 | 2020 | 322 | 322 | 322 |
| DE | NUTS 3 | 1999 | 2020 | 1678 | 1525 | 6558 |
| DK | NUTS 3 | 2006 | 2020 | 165 | 165 | 165 |
| EE | NUTS 3 | 2004 | 2020 | 81 | 81 | 81 |
| EL | NUTS 3 | 1998 | 2019 | 1047 | 1047 | 1047 |
| ES | NUTS 3 | 1998 | 2020 | 956 | 955 | 955 |
| FI | NUTS 3 | 1999 | 2020 | 328 | 328 | 328 |
| FR | NUTS 3 | 1989 | 2020 | 2993 | 2993 | 2993 |
| HR | NUTS 2 | 2008 | 2020 | 26 | 26 | 26 |
| HU | NUTS 3 | 1996 | 2020 | 492 | 492 | 492 |
| IE | - | - | - | - | - | - |
| IT | NUTS 3 | 2005 | 2020 | 1512 | 1512 | 1409 |
| LT | NUTS 3 | 2000 | 2020 | 208 | 208 | 210 |
| LU | NUTS 0 | 1975 | 2020 | 46 | 46 | 46 |
| LV | NUTS 3 | 1997 | 2018 | 110 | 110 | 110 |
| MT | NUTS 0 | 2000 | 2020 | 21 | 21 | 21 |
| NL | NUTS 2 | 1994 | 2020 | 324 | 324 | 324 |
| PL | NUTS 2 | 2003 | 2020 | 301 | 301 | 301 |
| PT | NUTS 2 | 1999 | 2020 | 154 | 147 | 147 |
| RO | - | - | - | - | - | - |
| SE | NUTS 3 | 1990 | 2020 | 628 | 628 | 628 |
| SI | - | - | - | - | - | - |
| SK | NUTS 3 | 2007 | 2018 | 96 | 96 | 96 |





**Table A7.** Number of records included in the dataset, length of time series and reported administrative level for crop *Grain maize*.

| Country | NUTS level | First year | Last year | # records Area | # records Production | # records Yield |
|---------|-----------|-----------|----------|---------------|---------------------|-----------------|
| AT | NUTS 2 | 1975 | 2020 | 414 | 414 | 414 |
| BE | NUTS 2 | 2011 | 2020 | 106 | 106 | 106 |
| BG | NUTS 2 | 1991 | 2020 | 180 | 180 | 180 |
| CY | NUTS 0 | 2000 | 2020 | 21 | 21 | 21 |
| CZ | NUTS 3 | 2005 | 2020 | 219 | 219 | 219 |
| DE | NUTS 1 | 2010 | 2020 | 149 | 131 | 131 |
| DK | NUTS 3 | 2011 | 2020 | 110 | 110 | 110 |
| EE | NUTS 3 | 2000 | 2020 | 105 | 105 | 105 |
| EL | NUTS 3 | 2009 | 2019 | 563 | 563 | 563 |
| ES | NUTS 3 | 1998 | 2020 | 1163 | 1157 | 1157 |
| FI | NUTS 3 | 2000 | 2020 | 399 | 399 | 399 |
| FR | NUTS 3 | 1989 | 2020 | 3026 | 3026 | 3026 |
| HR | NUTS 2 | 2005 | 2020 | 32 | 32 | 32 |
| HU | NUTS 3 | 1996 | 2020 | 492 | 492 | 492 |
| IE | NUTS 2 | 2000 | 2020 | 63 | 63 | 63 |
| IT | NUTS 3 | 1995 | 2020 | 2569 | 2566 | 2564 |
| LT | NUTS 3 | 2000 | 2020 | 209 | 209 | 210 |
| LU | NUTS 0 | 2000 | 2020 | 21 | 21 | 21 |
| LV | NUTS 3 | 2000 | 2020 | 126 | 126 | 126 |
| MT | NUTS 0 | 2000 | 2020 | 21 | 21 | 21 |
| NL | NUTS 2 | 2008 | 2020 | 156 | 156 | 156 |
| PL | NUTS 2 | 2003 | 2020 | 301 | 301 | 301 |
| PT | NUTS 2 | 1986 | 2020 | 245 | 245 | 245 |
| RO | NUTS 3 | 1998 | 2020 | 955 | 954 | 954 |
| SE | NUTS 3 | 2007 | 2020 | 10 | 10 | 10 |
| SI | NUTS 2 | 2007 | 2020 | 28 | 28 | 28 |
| SK | NUTS 3 | 2007 | 2018 | 94 | 94 | 94 |





**Table A8.** Number of records included in the dataset, length of time series and reported administrative level for crop *Sugar beet*.

| Country | NUTS level | First year | Last year | # records Area | # records Production | # records Yield |
|---|---|---|---|---|---|---|
| AT | NUTS 2 | 1975 | 2020 | 279 | 279 | 279 |
| BE | NUTS 2 | 1975 | 2020 | 468 | 470 | 441 |
| BG | NUTS 2 | 2008 | 2020 | 78 | 78 | 78 |
| CY | NUTS 0 | 2000 | 2020 | 21 | 21 | 21 |
| CZ | NUTS 3 | 1998 | 2020 | 270 | 270 | 270 |
| DE | NUTS 3 | 1999 | 2020 | 1733 | 1432 | 6051 |
| DK | NUTS 3 | 2006 | 2020 | 157 | 159 | 159 |
| EE | NUTS 3 | 1998 | 2020 | 115 | 115 | 115 |
| EL | NUTS 3 | 2009 | 2019 | 562 | 562 | 562 |
| ES | NUTS 3 | 1998 | 2020 | 490 | 486 | 486 |
| FI | NUTS 3 | 1998 | 2020 | 282 | 228 | 228 |
| FR | NUTS 3 | 1989 | 2020 | 2398 | 2397 | 2397 |
| HR | NUTS 2 | 2005 | 2020 | 18 | 18 | 18 |
| HU | NUTS 3 | 1996 | 2020 | 460 | 461 | 460 |
| IE | NUTS 2 | 2007 | 2020 | 42 | 42 | 42 |
| IT | NUTS 3 | 2006 | 2020 | 507 | 507 | 507 |
| LT | NUTS 3 | 1998 | 2020 | 186 | 184 | 184 |
| LU | NUTS 0 | 1975 | 2020 | 46 | 46 | 46 |
| LV | NUTS 3 | 2008 | 2020 | 78 | 78 | 78 |
| MT | NUTS 0 | 2000 | 2020 | 21 | 21 | 21 |
| NL | NUTS 2 | 1994 | 2020 | 324 | 324 | 324 |
| PL | NUTS 2 | 1999 | 2020 | 369 | 368 | 369 |
| PT | NUTS 2 | 1986 | 2020 | 245 | 245 | 245 |
| RO | NUTS 3 | 1990 | 2020 | 916 | 910 | 908 |
| SE | NUTS 3 | 1990 | 2020 | 132 | 132 | 132 |
| SI | NUTS 2 | 2007 | 2020 | 28 | 28 | 28 |
| SK | NUTS 3 | 1997 | 2018 | 120 | 120 | 141 |



**Table A9.** Number of records included in the dataset, length of time series and reported administrative level for crop *Sunflower*.

| Country | NUTS level | First year | Last year | # records Area | # records Production | # records Yield |
|---|---|---|---|---|---|---|
| AT | NUTS 2 | 1975 | 2020 | 311 | 311 | 310 |
| BE | NUTS 2 | 1975 | 2020 | 462 | 462 | 462 |
| BG | NUTS 2 | 1991 | 2020 | 180 | 180 | 180 |
| CY | NUTS 0 | 2000 | 2020 | 21 | 21 | 21 |
| CZ | NUTS 3 | 2005 | 2020 | 140 | 140 | 140 |
| DE | NUTS 1 | 1988 | 2020 | 405 | 271 | 257 |
| DK | NUTS 3 | 2000 | 2020 | 231 | 231 | 231 |
| EE | NUTS 3 | 2000 | 2020 | 105 | 105 | 105 |
| EL | NUTS 3 | 2009 | 2019 | 562 | 562 | 562 |
| ES | NUTS 3 | 1998 | 2020 | 877 | 875 | 874 |
| FI | NUTS 3 | 1990 | 2020 | 589 | 589 | 589 |
| FR | NUTS 3 | 1989 | 2020 | 2917 | 2916 | 2916 |
| HR | NUTS 2 | 2005 | 2019 | 20 | 20 | 20 |
| HU | NUTS 3 | 1996 | 2020 | 492 | 492 | 492 |
| IE | NUTS 2 | 1999 | 2020 | 60 | 66 | 60 |
| IT | NUTS 3 | 2006 | 2020 | 1063 | 1059 | 1059 |
| LT | NUTS 3 | 2000 | 2020 | 210 | 210 | 210 |
| LU | NUTS 0 | 2010 | 2020 | 11 | 11 | 11 |
| LV | NUTS 3 | 2000 | 2020 | 126 | 126 | 126 |
| MT | NUTS 0 | 2000 | 2020 | 21 | 21 | 21 |
| NL | NUTS 2 | 1975 | 2020 | 540 | 492 | 454 |
| PL | NUTS 2 | 1999 | 2020 | 372 | 372 | 368 |
| PT | NUTS 2 | 1986 | 2020 | 245 | 245 | 245 |
| RO | NUTS 3 | 1998 | 2020 | 891 | 890 | 890 |
| SE | NUTS 3 | 1975 | 2020 | 966 | 966 | 966 |
| SI | NUTS 2 | 2007 | 2020 | 28 | 28 | 28 |
| SK | NUTS 3 | 1995 | 2018 | 103 | 104 | 103 |





*Author contributions.* GR drafted the manuscript, created figures, processed and analyzed the data. GR, LNS, LS and IC collected data, performed data harmonization and defined post-processing rules. MvdV provided guidance and reviewed the manuscript. All the authors provided comments and suggestions on the paper.

*Competing interests.* The contact authors have declared that none of the authors has any competing interests.

*Acknowledgements.* The authors would like to acknowledge Steven Hoek and Hendrik Boogaard from Wageningen Environmental Research (WENR), and many colleagues, including Sara García-Condado, and Raul López-Lozano, who contributed to setting definitions and collecting data statistics over the years.



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
