# Peer review of "Harmonized European Union subnational crop statistics can reveal climate impacts and crop cultivation shifts"

_Earth System Science Data, 2023_

## Author Comment (AC1)

**Table 1.** Rules implemented for calculating new values and convert zero as null.

| Available variables: | | Yield to be calculated: | |
|---|---|---|---|
| AREA | PRODUCTION | *0* | *NA* |
| | *0* | 0 | 0** |
| *0* | *NA* | 0 | NA |
| | *Value* | NA* | NA |
| | *0* | 0 | NA |
| *NA* | *NA* | NA* | NA |
| | *Value* | NA* | NA |
| | *0* | NA | NA |
| *Value* | *NA* | NA | NA |
| | *Value* | P/A** | P/A** |

| Available variables: | | Production to be calculated: | |
|---|---|---|---|
| AREA | YIELD | *0* | *NA* |
| | *0* | 0 | 0** |
| *0* | *NA* | 0 | NA |
| | *Value* | NA* | NA |
| | *0* | 0 | NA |
| *NA* | *NA* | NA* | NA |
| | *Value* | NA* | NA |
| | *0* | NA | NA |
| *Value* | *NA* | NA | NA |
| | *Value* | AxY** | AxY** |

| Available variables: | | Area to be calculated: | |
|---|---|---|---|
| YIELD | PRODUCTION | *0* | *NA* |
| | *0* | 0 | 0** |
| *0* | *NA* | 0 | NA |
| | *Value* | NA* | NA |
| | *0* | 0 | NA |
| *NA* | *NA* | NA* | NA |
| | *Value* | NA* | NA |
| | *0* | NA | NA |
| *Value* | *NA* | NA | NA |
| | *Value* | P/Y** | P/Y** |

* ZERO_AS_NULL = Yes; ** CALCULATED_V = Yes; NA = Not Available

**Table 2.** Rules implemented for calculating new values for total wheat or total barley.

| AREA | | |
|---|---|---|
| Crop1 | Crop2 | Total |
| *0* | *0* | 0* |
| | *NA* | NA |
| | *Value* | $A_{crop1} + A_{crop2}$* |
| *NA* | *0* | NA |
| | *NA* | NA |
| | *Value* | NA |
| *Value* | *0* | $A_{crop1} + A_{crop2}$* |
| | *NA* | NA |
| | *Value* | $A_{crop1} + A_{crop2}$* |

| PRODUCTION | | |
|---|---|---|
| Crop1 | Crop2 | Total |
| *0* | *0* | 0* |
| | *NA* | NA |
| | *Value* | $P_{crop1} + P_{crop2}$* |
| *NA* | *0* | NA |
| | *NA* | NA |
| | *Value* | NA |
| *Value* | *0* | $P_{crop1} + P_{crop2}$* |
| | *NA* | NA |
| | *Value* | $P_{crop1} + P_{crop2}$* |

| YIELD | | |
|---|---|---|
| Area $_{total}$ | Production $_{total}$ | Yield $_{total}$ |
| *0* | *0* | 0* |
| | *NA* | NA |
| | *Value* | NA |
| *NA* | *0* | NA |
| | *NA* | NA |
| | *Value* | NA |
| *Value* | *0* | $P_{total}/A_{total}$* |
| | *NA* | $[(A_{crop1} x Y_{crop1})+(A_{crop2} x Y_{crop2})]/A_{total}$* |
| | *Value* | $P_{total}/A_{total}$* |

* CALCULATED_V = Yes; NA = Not Available

**Table 3.** Rules implemented for checking agreement among values of Area, Production and Yield.

| COHERENCE_APY | | |
|---|---|---|
| Yes | No | NA |
| abs(P-(AxY)) <= 0.01xP | abs(P-(AxY)) > 0.01xP | Any variable is NA |

**Table 4.** Rules implemented for checking agreement among values of Total wheat (Total barley), Soft wheat (Winter barley), and Durum wheat (Spring barley).

| COHERENCE_CROP - Area | | |
|---|---|---|
| Yes | No | NA |
| $abs(A_{total} -(A_{crop1} + A_{crop2}) <= 0.01xA_{total}$ | $abs(A_{total}-(A_{crop1} + A_{crop2})) > 0.01xA_{total}$ | Any crop is NA |

| COHERENCE_CROP - Production | | |
|---|---|---|
| Yes | No | NA |
| $abs(P_{total} -(P_{crop1} + P_{crop2})) <= 0.01xP_{total}$ | $abs(P_{total} -(P_{crop1} + P_{crop2})) > 0.01xP_{total}$ | Any crop is NA |

| COHERENCE_CROP - Yield | | |
|---|---|---|
| Yes | No | NA |
| $abs(Y_{total} - [(Y_{crop1}xA_{crop1} +Y_{crop2}xA_{crop2})/ A_{total}]) <= 0.01xY_{total}$ | $abs(Y_{total} - [(Y_{crop1}xA_{crop1} +Y_{crop2}xA_{crop2})/ A_{total}]) > 0.01xY_{total}$ | Any crop is NA |

---

## Author Comment (AC2)

Czechia:

| Region | Average production [1000 t] (Reference period: 2000 – 2020) | | | | | |
|--------|-----------|--------------|---------------|-------------|-----------|-----------|
|        | Soft wheat | Winter barley | Spring barley | Grain maize | Sunflower | Sugar beet |
| CZ010 | 25.4 | 1.2 | 8207.7 | 1.7 | 0 | 19.1 |
| CZ020 | 945.5 | 111.5 | 257.3 | 110.4 | 12.7 | 977.3 |
| CZ031 | 410.5 | 79 | 96.5 | 23.7 | 0.1 | 0 |
| CZ032 | 324.1 | 85.2 | 63.5 | 14.6 | 1.4 | 0 |
| CZ041 | 31.4 | 7.5 | 15.6 | 0.2 | 0 | 0 |
| CZ042 | 346.6 | 24.6 | 93.5 | 27.7 | 6.2 | 194.7 |
| CZ051 | 62.4 | 9.7 | 15.6 | 2 | 0 | 35.9 |
| CZ052 | 307.3 | 43.8 | 48.7 | 41.3 | 0.6 | 586.9 |
| CZ053 | 273.5 | 33.2 | 80.6 | 56.9 | 3.2 | 230.6 |
| CZ063 | 362.6 | 56.7 | 172.4 | 20.3 | 0.3 | 14.2 |
| CZ064 | 588.9 | 34.2 | 185.9 | 292.3 | 27.9 | 320.3 |
| CZ071 | 304.2 | 16.6 | 176.4 | 57.1 | 0.3 | 695.5 |
| CZ072 | 184.2 | 14.1 | 49.2 | 54.4 | 2 | 109.2 |
| CZ080 | 208.6 | 26.8 | 65.8 | 30.3 | 0.3 | 367.1 |
| CZ | 4405 | 544.1 | 1329.3 | 733 | 550.9 | 3550.8 |

[Figure]

Bulgaria:

| Region | Average production [1000 t] (Reference period: 2000 – 2020) | | | | |
|---|---|---|---|---|---|
| | Soft wheat | Durum wheat | Total barley | Grain maize | Sunflower |
| BG31 | 1111.7 | 1.5 | 107.1 | 584.6 | 310.2 |
| BG32 | 1140.6 | 0.5 | 169.9 | 535.4 | 320.0 |
| BG33 | 1352.2 | 7.7 | 157.7 | 713.4 | 365.9 |
| BG34 | 941.9 | 29.9 | 201.7 | 37.1 | 190.2 |
| BG41 | 150.6 | 2 | 56.9 | 48.9 | 54.6 |
| BG42 | 397.9 | 19.9 | 38.2 | 40.6 | 81.2 |
| BG | 5095.1 | 61.4 | 731.7 | 1959.9 | 1322.1 |

---

## Author Response (AR1)

European Commission
Joint Research Centre
Directorate for Sustainable Resources
Food Security – D5
Via Fermi 2749, TP 272, 27b/019
I-21027 ISPRA (VA), ITALY

January 12th, 2024

Dear Editor,

Please find enclosed our point-by-point reply to referees' comments. We addressed all the comments kindly pointed out by the reviewers and accordingly proposed improvements to the manuscript.

In our responses to referees, our replies are in red.

We would like to thank the two anonymous referees for finding the time to provide helpful suggestions on our manuscript.

Sincerely on behalf of all authors,

Giulia Ronchetti and Marijn van der Velde

**Anonymous Referee #1**

The manuscript presents a harmonized dataset of subnational crop statistics, encompassing crop area, production, and yields for European countries. Spanning a median period of 21 years, from 1975 to 2020, this dataset is compiled from input data sourced from National Statistical Offices (NSIs) and country-reported data to Eurostat. The latter source provides data at both subnational and national levels. To ensure consistency, the authors harmonized the input data to address variations in crop definitions, reporting methods (such as combining totals for cereals versus separate data for winter and spring cereals), units, and changes in administrative reporting units. Data gaps were filled using simple imputation rules. Additionally, the manuscript includes examples illustrating potential applications of the dataset.

While the methodology used isn't groundbreaking, the dataset could serve as a valuable reference for studies investigating cropping trends across Europe. However, there are several concerns I recommend the authors address before final acceptance for publication:

Reply: We thank the reviewer for their assessment and provide a point-by-point reply in response to the concerns raised.

1. The introductory section and the paragraph discussing "Outlook and recommendations" seem disjointed. Consider beginning the manuscript by introducing the SAIO regulation in the European context, emphasizing the dataset's potential and the lessons learned from its development for this directive. Highlight how this dataset might expand to include agricultural inputs, aligning with the SAIO regulation's primary focus.

Reply: We agree that the two sections seem a bit disjointed, and we decided to mention SAIO in the introduction. However, when rewriting the paper, it actually flows better if we maintain the Outlook and Recommendations part towards the end, which avoids coming up with the recommendations in the introduction. In addition, SAIO does not apply to past statistics (included in this dataset) but will be in place from 1st January 2025.
Revised lines 45 – 51: *Yet, despite their importance, a complete and harmonized collection of subnational crop statistics for countries in the European Union (EU) currently does not exist. A new EU framework regulation governing the collection of Statistics on Agricultural Inputs and Outputs (SAIO) will apply from 1st January 2025. The collection of subnational crop statistics will become legally binding (European Parliament, EPRS). This new regulation should improve the availability and quality of subnational statistics at EU-level considerably. Currently, Eurostat (Eurostat, 2020) receives subnational statistics from Member States (MS) and reports these in the regional database (Eurostat, a), the data provision relies on voluntary contributions, contains gaps, and does not consider changes in regional administrative boundaries through time.*

2. Yields should be computed from harvested area rather than planted (sown) area. This distinction could significantly impact the coherence analyzed in the manuscript. The authors should discuss this aspect of the input data explicitly, as there seems to be a generic reference to crop area throughout the text. The metadata file, specifically the "Summary of algorithm for disaggregation," lacks clarity on this distinction. Moreover, the manuscript lacks reporting on limitations and caveats.

Reply: While we agree with this assessment, the nature of statistical collection often results in sown area being reported as it is derived from aggregated area values from the farmers' applications for the Common Agricultural Policy that are done at the start of the season and not updated. However, it is not always known if area values refer to sown or harvested area. Such information is especially

important during heavy impacts where sown area may have been strongly reduced. These caveats are now discussed in the manuscript.

New paragraph added in section 3.2, lines 289 - 294: *Main causes that can affect coherence among variables are to be attributed to the distinction between sown and harvested area. It is not always known if area values refer to sown or harvested area, since only a few statistical sources either publish both values or clearly mention values they refer to. The nature of statistical collection often results in sown area being reported, as it is derived from aggregated area values from the farmers' applications for the Common Agricultural Policy that are done at the start of the season and not updated. However, such information is especially important during heavy impacts where sown area may have been strongly reduced and this can result in inconsistencies with production and yield data.*

3. Consolidate the imputation rules into a simple table. Report the total number of inputted data; does this align with the number of records marked CALCULATED_V=Yes (approximately 125k or 37% of total records)? However, there's confusion: records with both null values and columns ZERO_AS_NULL and CALCULATED_V marked as "Yes" seem contradictory.

Reply: Input data amounts to 219472 values, final dataset consists in 344282 values, so 124810 new records have been calculated (and marked as CALCULATED_V = Yes). We have included the number of input records in the revised manuscript (line 96 - 97): *The latest data were collected in July 2022, accounting to 219472 records for the crops included in this dataset.*

Among 124810 calculated values, 22 records report also ZERO_AS_NULL flag. This means that these records were missing from input data and were calculated in the first step of post-processing phase, with computed value equal to zero. Then, these values were turned into null since they caused inconsistencies with the other variables (as explained in section 2.3.2). As shown in the workflow in Figure 1, post-processing steps are part of a cascade procedure that generated both flags.

In addition, we have consolidated imputation rules into a simple table, as suggested, and can include them in the ancillary documentation.

**Table 1.** Rules implemented for calculating new values and convert zero as null.

| Available variables: | | Yield to be calculated: | |
|---|---|---|---|
| AREA | PRODUCTION | *0* | *NA* |
| *0* | *0* | 0 | 0** |
| | *NA* | 0 | NA |
| | *Value* | NA* | NA |
| *NA* | *0* | 0 | NA |
| | *NA* | NA* | NA |
| | *Value* | NA* | NA |
| *Value* | *0* | NA | NA |
| | *NA* | NA | NA |
| | *Value* | P/A** | P/A** |

| Available variables: | | Production to be calculated: | |
|---|---|---|---|
| AREA | YIELD | *0* | *NA* |
| *0* | *0* | 0 | 0** |
| | *NA* | 0 | NA |
| | *Value* | NA* | NA |
| *NA* | *0* | 0 | NA |
| | *NA* | NA* | NA |
| | *Value* | NA* | NA |
| *Value* | *0* | NA | NA |
| | *NA* | NA | NA |
| | *Value* | AxY** | AxY** |

| Available variables: | | Area to be calculated: | |
|---|---|---|---|
| YIELD | PRODUCTION | *0* | *NA* |
| *0* | *0* | 0 | 0** |
| | *NA* | 0 | NA |
| | *Value* | NA* | NA |
| *NA* | *0* | 0 | NA |
| | *NA* | NA* | NA |
| | *Value* | NA* | NA |
| *Value* | *0* | NA | NA |
| | *NA* | NA | NA |
| | *Value* | P/Y** | P/Y** |

* ZERO_AS_NULL = Yes; ** CALCULATED_V = Yes; NA = Not Available

**Table 2.** Rules implemented for calculating new values for total wheat or total barley.

| AREA | | |
|---|---|---|
| Crop1 | Crop2 | Total |
| *0* | *0* | 0* |
| | *NA* | NA |
| | *Value* | $A_{crop1} + A_{crop2}$* |
| *NA* | *0* | NA |
| | *NA* | NA |
| | *Value* | NA |
| *Value* | *0* | $A_{crop1} + A_{crop2}$* |
| | *NA* | NA |
| | *Value* | $A_{crop1} + A_{crop2}$* |

| PRODUCTION | | |
|---|---|---|
| Crop1 | Crop2 | Total |
| *0* | *0* | 0* |
| | *NA* | NA |
| | *Value* | $P_{crop1} + P_{crop2}$* |
| *NA* | *0* | NA |
| | *NA* | NA |
| | *Value* | NA |
| *Value* | *0* | $P_{crop1} + P_{crop2}$* |
| | *NA* | NA |
| | *Value* | $P_{crop1} + P_{crop2}$* |

| YIELD | | |
|---|---|---|
| Area $_{total}$ | Production $_{total}$ | Yield $_{total}$ |
| *0* | *0* | 0* |
| | *NA* | NA |
| | *Value* | NA |
| *NA* | *0* | NA |
| | *NA* | NA |
| | *Value* | NA |
| *Value* | *0* | $P_{total}/A_{total}$* |
| | *NA* | $[(A_{crop1}xY_{crop1})+(A_{crop2}xY_{crop2})]/A_{total}$* |
| | *Value* | $P_{total}/A_{total}$* |

* CALCULATED_V = Yes; NA = Not Available

**Table 3.** Rules implemented for checking agreement among values of Area, Production and Yield.

| COHERENCE_APY | | |
|---|---|---|
| Yes | No | NA |
| $abs(P-(A \times Y)) <= 0.01 \times P$ | $abs(P-(A \times Y)) > 0.01 \times P$ | Any variable is NA |

**Table 4.** Rules implemented for checking agreement among values of Total wheat (Total barley), Soft wheat (Winter barley), and Durum wheat (Spring barley).

| COHERENCE_CROP - Area | | |
|---|---|---|
| Yes | No | NA |
| $abs(A_{total} -(A_{crop1} + A_{crop2}) <= 0.01 \times A_{total}$ | $abs(A_{total}-(A_{crop1} + A_{crop2})) > 0.01 \times A_{total}$ | Any crop is NA |

| COHERENCE_CROP - Production | | |
|---|---|---|
| Yes | No | NA |
| $abs(P_{total} -(P_{crop1} + P_{crop2})) <= 0.01 \times P_{total}$ | $abs(P_{total} -(P_{crop1} + P_{crop2})) > 0.01 \times P_{total}$ | Any crop is NA |

| COHERENCE_CROP - Yield | | |
|---|---|---|
| Yes | No | NA |
| $abs(Y_{total} - [(Y_{crop1} \times A_{crop1} +Y_{crop2} \times A_{crop2})/ A_{total}]) <= 0.01 \times Y_{total}$ | $abs(Y_{total} - [(Y_{crop1} \times A_{crop1} +Y_{crop2} \times A_{crop2})/ A_{total}]) > 0.01 \times Y_{total}$ | Any crop is NA |

4. Consider restructuring the flagging system into one field detailing official and estimated values, providing details on the source type or the method used for imputation/estimation.

Reply: We refrain from doing this as it is our assessment that it is clearer to keep the organization of the flags as currently proposed.

Line 395 correct typo: yiled.

Reply: Done. We thank the referee for noticing this typo.

**Anonymous Referee #2**

This study developed a harmonized dataset of subnational crop statistics, including crop area, production, and yields for the European Union (EU). Such benchmark data will be helpful to better analyze the past, understand the present, and predict future trends in yield, area, and production. The descriptions of data collection, data harmonization, and post processing are very detailed, though the method is simple. However, the manuscript is not well organized. It provides too much metadata information about the data, which makes it look like a technical report rather than a scientific article. Moreover, there is no overall data analysis and assessment. Thus, I think this manuscript needs to be much improved and can't be published in ESSD.

Reply: While we thank the referee for their assessment that this benchmark data will be helpful, we do not agree with the final assessment that this manuscript cannot be published in ESSD. In this paper we present a unique dataset of European crop statistics: we believe we have done this in a structured way and believe that all information (metadata) provided is essential to understand the specific challenges of this exercise (bringing together such a variety of national datasets) and to reuse the proposed dataset.
In addition, the referee fails to provide practical recommendations on how to improve organization. We are of the opinion that we have provided an analysis of the data, and as such contest that there is no overall data analysis. To illustrate the value of the dataset, we provide straightforward data exploration with two examples. Undoubtedly, further data analysis and reuse of the data by others, and combinations with other datasets, will proof the value of the dataset.

Specific comments:

1. The title of this manuscript, "Harmonized European Union subnational crop statistics reveal climate impacts and crop cultivation shifts" includes information about the impacts of climate change and crop cultivation shifts in the EU. However, the manuscript only discussed a little bit about the crop production shift and variations. If the manuscript continues to use the title, more analysis about how climate change impacts crop diversity, crop planted area, and crop production should be included in the data analysis part.

Reply: We agree that the title of the manuscript is a bit ambitious, but our intention was to emphasize the potential of such a harmonized statistical dataset. We propose a new title: "Harmonized European Union subnational crop statistics can reveal climate impacts and crop cultivation shifts". However, we would like to point out that in our analysis we provide some evidence of climate impacts on crop cultivation shift, but this is not the only affecting factor, as also economic and political decision may have an impact on crops shifts. Our harmonized statistical dataset includes long time-series that can allow to perform specific analyses according to users' needs and can serve as reference dataset for crop modelling and agro-climate studies.

2. The newly developed dataset has seven crops, including soft/durum wheat, winter/spring barley, grain maize, sugar beet, and sunflower. I wonder how much of these primary crops accounted for all the crop area in the EU?

Reply: We have calculated these numbers and included the following paragraph in the manuscript (lines 78 – 79): *According to Eurostat data, in the last three years these crops jointly accounted for 50,7% of the arable land in the EU (Eurostat, a).*

3. The introduction mainly focuses on the weakness of the currently available datasets (e.g., Eurostat, NSIs). The author can add some research needs in other fields for such datasets, such as the time-series analysis of crop area and production, to be consistent with the discussion.

Reply: The introduction is now improved by mentioning the upcoming SAIO regulation. Also, to better motivate the study, we now emphasize the use of such time-series in combination with crop model and Earth Observation assessments. Lines 25 – 33: *Among the main application fields that rely on subnational crop statistics, remote sensing-based analysis and crop model estimations play a major role. To name a few, Blickensdörfer et al. make use of area crop statistics as independent dataset to assess the accuracy of crop type maps derived from a combination of satellite imagery; similarly, in d'Andrimont et al. (2021) and Becker-Reshef et al. (2023) crop maps are compared to official subnational statistics to validate results. The major global agricultural monitoring systems (Fritz et al., 2019) rely on well consolidated time series of crop statistics to develop estimation models for providing crop yield forecasts (Schauberger et al., 2020), but many other studies are available in the literature where crop statistical datasets are used to calibrate and validate crop models performances from local to global level (Paudel et al., 2022; Kern et al., 2018; Neumann and Smith, 2018; Kowalik et al., 2014).*

4. The discussion part (section 4), especially sections 4.1.1 and 4.12, includes both data analysis and discussion, which makes me lost. As suggested in comment 1, the data analysis should be added to the results part.

Reply: The scope of this manuscript is to present and detail a statistical dataset. The Discussion part reports potential uses of the proposed dataset and some lessons learned during the collection and preparation of it. To illustrate the value of the dataset, we provide straightforward data exploration with two examples, but we would not limit the use of this dataset to these and leave to users the possibility to manipulate and adapt the dataset at their needs. Therefore, we prefer to keep Discussion section in its current version, to be considered as suggestions for further applications of the dataset.

5. Before discussing the potential use of this dataset, a data comparison with other datasets (e.g., remote sensing-based cropland area) should be conducted to prove the advances of the new dataset.

Reply: We rather believe that the statistical dataset we present here can become a benchmark for multi-annual remote sensing-based assessments and can be used for bias assessment as well. As an example, in d'Andrimont, et al., 2021, authors used Earth Observation data to map crop area across the EU for 2018 and then used subnational statistics for a single year to compare to this assessment. We do not see the need to do such comparisons in the present paper, but in the revised manuscript we have included a paragraph about the current use of subnational crop statistics in literature, to enhance the need of such a consistent dataset. Lines 25 – 33: *Among the main application fields that rely on subnational crop statistics, remote sensing-based analysis and crop model estimations play a major role. To name a few, Blickensdörfer et al. make use of area crop statistics as independent dataset to assess the accuracy of crop type maps derived from a combination of satellite imagery; similarly, in d'Andrimont et al. (2021) and Becker-Reshef et al. (2023) crop maps are compared to official subnational statistics to validate results. The major global agricultural monitoring systems (Fritz et al., 2019) rely on well consolidated time series of crop statistics to develop estimation models for providing crop yield forecasts (Schauberger et al., 2020), but many other studies are available in the literature where crop statistical datasets are used to calibrate and validate crop models performances from local to global level (Paudel et al., 2022; Kern et al., 2018; Neumann and Smith, 2018; Kowalik et al., 2014).*

Reference:
d'Andrimont, R., Verhegghen, A., Lemoine, G., Kempeneers, P., Meroni, M. and Van der Velde, M., 2021. From parcel to continental scale–A first European crop type map based on Sentinel-1 and LUCAS Copernicus in-situ observations. *Remote sensing of environment*, 266, p.112708.

6. Line 295: As you said, crop statistics in this dataset cover a long time range, from 1975 to 2020, to allow time series analysis. However, there is no spatial and temporal trend analysis for crop production or crop area.

Reply: We believe that such analysis can subsequently be done, also with respect to the specific context any users may have in mind. With this paper, our purpose was not to give a full exploration of the data but to provide a consistent data collection. We are confident that spatial and temporal trend analysis will be part of further studies from us or other authors, as the effort needed to collect and homogenize a large dataset is already covered by this publication.

7. Line 300-301: Why were the lowest yield values registered at the beginning of the time series, mostly in the early 2000s? Was it because of drought and heatwave? Please give some explanations.

Reply: The main reason why the lowest yield values were registered at the beginning of time-series is that yields tend to gradually increase over time, particularly in countries where agronomic techniques and management practices significantly improved in recent years. The effects of such improvements usually are reflected in the time trend component of the yield series, therefore a proper detrending algorithm should be applied to detect this. However, we did not detrend yields, as this was not the scope of our analysis. We wanted to illustrate the value of the dataset and do not provide any suggestions about the trend approach. In fact, to properly do such an analysis is not that straightforward and many different strategies can be applied, which we now refer to (Michel and Makowski, 2013; Schauberger et al., 2018).
Reference:
Michel, L. and Makowski, D., 2013. Comparison of statistical models for analyzing wheat yield time series. *PLoS One*, 8(10), p.e78615.
Schauberger, B., Ben-Ari, T., Makowski, D., Kato, T., Kato, H. and Ciais, P., 2018. Yield trends, variability and stagnation analysis of major crops in France over more than a century. *Scientific reports*, 8(1), p.16865.

8. Line 302-304: How do you conclude that improvements in agronomic techniques and management practices have less impact on crop yields? Please add some references or supporting figures.

Reply: The meaning behind this sentence relates to the fact that trends are associated to such improvements. In fact, the variability of yield values can be explained by factors related to agronomic technologies and weather. According to literature (Ceglar et al., 2016; Garcia-Condado et al., 2019), the impact of management improvements (cultivars, practices, technological advances, ...) are reflected in the time trend component of yield series, while weather changes are the main causes of inter-annual yield variability. In eastern and north-eastern European countries (e.g., Bulgaria, Romania, Hungary, Slovakia, Poland, Lithuania) agronomic techniques and management practices are still improving, therefore in this country we can observe a high trend effect on yield values. Conversely, in western countries (e.g., France, Germany, Italy), advances in agricultural technologies have stopped, so that the yield trend component is neglectable, and yield changes can be attributed to weather/climate factors. In the revised manuscript, we refer to a recent study that evaluated yield trend component in European countries (Ronchetti et al., 2023).

Reference:

Ceglar, A., Toreti, A., Lecerf, R., Van der Velde, M. and Dentener, F., 2016. Impact of meteorological drivers on regional inter-annual crop yield variability in France. *Agricultural and Forest Meteorology*, 216, pp.58-67.

García-Condado, S., López-Lozano, R., Panarello, L., Cerrani, I., Nisini, L., Zucchini, A., Van der Velde, M. and Baruth, B., 2019. Assessing lignocellulosic biomass production from crop residues in the European Union: Modelling, analysis of the current scenario and drivers of interannual variability. *GCB Bioenergy*, 11(6), pp.809-831.

Ronchetti, G., Manfron, G., Weissteiner, C.J., Seguini, L., Scacchiafichi, L.N., Panarello, L. and Baruth, B., 2023. Remote sensing crop group-specific indicators to support regional yield forecasting in Europe. *Computers and Electronics in Agriculture*, 205, p.107633.

9. Line 305. What's the meaning of "trend effect"?

Reply: Please, consider our reply to the previous point.

10. Line 309-313: When giving an example about climate change resulting in crop yield loss, it would be better to show the annual precipitation or drought index rather than cite a paper only.

Reply: In this context, we prefer to cite papers rather than show a meteorological index. Crop yield losses mentioned in the manuscript were caused by a combination of factors, mostly climatic, and the cited articles deeply analyse all of them.

11. Line 312-317: These descriptions are not correlated with the crop production time-series analysis.

Reply: We kindly ask the referee to better clarify this comment. As already stated, we have proposed some analysis to evidence the potential of our dataset. In this case, we studied lowest and highest yielding years by means of the statistical dataset and discussed results. Our intention is to provide some application examples and not a full analysis.

12. Line 328-330: I think we can't get such a conclusion from Figure 7. Some supporting figures should be given, such as a histogram of the subnational crop yield.

Reply: For the sake of clarity, in Figure 7 we represent crop production values, therefore histograms of subnational crop yield cannot directly support the interpretation of the Figure. However, histograms of subnational crop production will help in better understanding proposed maps. Due to space limitations, we cannot include such histograms in the manuscript, but we refer to Supplementary Materials, where readers can already find tables in excel format reporting descriptive statistics summarizing subnational crop statistics. These tables can be easily used to generate subnational crop production histograms.

To better understand our statement, we report here a map representing geographic centroid and crop production centroids for Czechia and a table summarizing subnational production values and we refer to our reply to comment 13 with a description of the computation of crop production centroids.

In the map, crop production centroids spread around the geometric centroid of the country, as every NUTS unit contributes to national crop production. For some crops, production centroids slightly move away from the geographic centroid since there are some NUTS regions with greater production. These are the cases of soft wheat and winter barley, due to an increased production in CZ020, and grain maize, where we observe a peak of production in CZ064.

We revised the manuscript accordingly (lines 344 - 349): *The resulting map represents the spatial distribution of the production for each crop in the different countries. In some countries, including central and northeastern Europe (e.g. Czechia, Slovakia, Poland, Lithuania, Latvia, Estonia) production centroids for the different crops are located in the same area, roughly corresponding to the geometric centroids of the country. This can suggest that all subnational units almost equally*

*concur to national crop production and there are few regions providing a greater contribution (Joint Research Centre, 2023; Lennert and Farkas, 2020; Rega et al., 2020; Lopez-Lozano et al., 2015).*

| Region | Average production [1000 t] (Reference period: 2000 – 2020) | | | | | |
|---|---|---|---|---|---|---|
| | Soft wheat | Winter barley | Spring barley | Grain maize | Sunflower | Sugar beet |
| CZ010 | 25.4 | 1.2 | 8207.7 | 1.7 | 0 | 19.1 |
| CZ020 | 945.5 | 111.5 | 257.3 | 110.4 | 12.7 | 977.3 |
| CZ031 | 410.5 | 79 | 96.5 | 23.7 | 0.1 | 0 |
| CZ032 | 324.1 | 85.2 | 63.5 | 14.6 | 1.4 | 0 |
| CZ041 | 31.4 | 7.5 | 15.6 | 0.2 | 0 | 0 |
| CZ042 | 346.6 | 24.6 | 93.5 | 27.7 | 6.2 | 194.7 |
| CZ051 | 62.4 | 9.7 | 15.6 | 2 | 0 | 35.9 |
| CZ052 | 307.3 | 43.8 | 48.7 | 41.3 | 0.6 | 586.9 |
| CZ053 | 273.5 | 33.2 | 80.6 | 56.9 | 3.2 | 230.6 |
| CZ063 | 362.6 | 56.7 | 172.4 | 20.3 | 0.3 | 14.2 |
| CZ064 | 588.9 | 34.2 | 185.9 | 292.3 | 27.9 | 320.3 |
| CZ071 | 304.2 | 16.6 | 176.4 | 57.1 | 0.3 | 695.5 |
| CZ072 | 184.2 | 14.1 | 49.2 | 54.4 | 2 | 109.2 |
| CZ080 | 208.6 | 26.8 | 65.8 | 30.3 | 0.3 | 367.1 |
| CZ | 4405 | 544.1 | 1329.3 | 733 | 550.9 | 3550.8 |

13. Figure 7 shows the spatial representation of crop production centroids within each country in the time period 2000-2020. How do you calculate the centroids? Could you please provide an equation? When calculating the centroids, do you use the crop statistics data at the same NUT level? If you don't use the same NUT level, the conclusion about the spatial shift of crop production would be biased because the mean area of administrative regions at NUT3 is significantly smaller than NUT2, such as Germany.

Reply: Centroids are georeferenced points representing the geographic centre of a region/spatial unit. In this manuscript, we have extended the concept of centroids to crop production centroids, by means of weights based on crop production values. The result represents the geographic distribution of crop production in each country. To compute crop production centroids, we first selected a unique NUTS level in each country (see Tables A1-A9 in Appendix A), then we calculated the geographic centroid of each NUTS unit and finally we summed up all NUTS centroids using crop production as weights (see equation below). The mean area of administrative units is not affecting the computation, as it is not part of the equation.

$$X_{Pj} = \sum_{i=1}^{n} \frac{X_i * P_{ij}}{P_{ij}}$$

$$Y_{Pj} = \sum_{i=1}^{n} \frac{Y_i * P_{ij}}{P_{ij}}$$

where $X_{Pj}$ and $Y_{Pj}$ represent eastern and northern coordinates of the crop production centroid for crop $j$, $X_i$ and $Y_i$ represent eastern and northern coordinates of the geographic centroid for NUTS region $i$, $P_{ij}$ represent average production for crop $j$ in NUTS region $i$.

As an example, we provide here a map representing geographic centroid and crop production centroids for Bulgaria and a table summarizing subnational production values. Geographic centroid lays in the middle of the country, while crop production centroids follow subnational crop production distribution. Particularly, grain maize centroid collocates in the north half of the country, since northern regions (i.e., BG31, BG32 and BG33) dominate grain maize production, while durum wheat centroid moves to the south, as the most production is concentrated in southern regions (i.e., BG34 and BG42).

| Region | Average production [1000 t] (Reference period: 2000 – 2020) | | | | |
|---|---|---|---|---|---|
| | Soft wheat | Durum wheat | Total barley | Grain maize | Sunflower |
| BG31 | 1111.7 | 1.5 | 107.1 | 584.6 | 310.2 |
| BG32 | 1140.6 | 0.5 | 169.9 | 535.4 | 320.0 |
| BG33 | 1352.2 | 7.7 | 157.7 | 713.4 | 365.9 |
| BG34 | 941.9 | 29.9 | 201.7 | 37.1 | 190.2 |
| BG41 | 150.6 | 2 | 56.9 | 48.9 | 54.6 |
| BG42 | 397.9 | 19.9 | 38.2 | 40.6 | 81.2 |
| BG | 5095.1 | 61.4 | 731.7 | 1959.9 | 1322.1 |

[Figure]

| | | |
|---|---|---|
| ■ Geographic | ■ Durum wheat | ■ Grain maize |
| ■ Soft wheat | ■ Total barley | ■ Sunflower |

14. According to Figure 7, there was a spatial shift of the centroid of crop production. The authors explained it was induced by climate change by citing Ceglar et al. (2019). Does the centroid of the crop area also show the same spatial shift? Maybe it is more sensitive than the crop production.

Reply: We suppose that the referee was referring to Figure 8. In our assessment, we decided to show shifts in crop production as these are more sensitive, including factors that can affect both area and yield values. On the one hand, impacts on crop area may lead to the introduction of new crops in some regions, on the other hand, impacts on crop yield may result in abrupt increase/decrease in crop production. Both impacts can be depicted by analysing effects on crop production. Moreover, area changes can also be attributed to the impact of agricultural policies. Market needs, agro-economic regulations, agricultural support policy are all factors contributing to the variability of crop area.

15. Line 364-365: Ceglar et al. (2019) found that agro-climate zones in eastern rather than southern Europe have experienced a northward migration velocity of 100 km per 10 years over the past 40 years. In addition, their study period is 1975-1995 and 1996-2016. It doesn't support the current conclusion.

Reply: In our analysis, we do not provide an exercise as in Ceglar et. al., (2019) but we provide evidence of the usefulness of our harmonised statistical dataset that is in line with Ceglar et. al., (2019). In the cited paper, authors report analysis for period 1975-2016, but also report projections for the coming decades. With their analysis, a northward shift of agro-climate zones in Europe can be observed and, according to their projections, these shifts may also accelerate particularly in Eastern Europe. To perform such analysis, authors made use of long series of agro-meteorological indicators and accurate models. To be noticed that in our analysis we make use of crop statistics that importantly, also encompass production shifts due to non-climate factors, such as economic and political decisions. To better clarify this, in the revised manuscript we rephrase our final statements.

Lines 383 – 388: *Ceglar et al. observed a migration velocity of agro-climate zones northward of 100 km per 10 years solely using agro-metereological indicators, while we find results comparable in magnitude through the analysis of crop statistics for grain maize and durum wheat. In addition, we show production shifts not mediated by climate but rather by economic opportunity, largely in eastern Europe, as illustrated by the high eastward shift rate of sunflower production. Hence, the completeness of subnational crop statistics presented here can be of help to reveal changes in agricultural cultivation zones due to various concurring factors with subsequent crop production impacts.*

---

## Author Response (AR2)

European Commission
Joint Research Centre
Directorate for Sustainable Resources
Food Security – D5
Via Fermi 2749, TP 272, 27b/019
I-21027 ISPRA (VA), ITALY

February 02nd, 2024

Dear Editor,

We are pleased to provide the revised version of our manuscript titled "*Harmonized European Union subnational crop statistics can reveal climate impacts and crop cultivation shifts*".

According to your suggestion, in this version we have just adjusted the share of arable land covered by the crops included in this study.

We would like to thank you again for your assessment and valuable comment on our data paper.

Sincerely on behalf of all authors,

Giulia Ronchetti and Marijn van der Velde